



# Stability assessment of degrading permafrost rock slopes based on a coupled thermo-mechanical model

Philipp Mamot, Samuel Weber, Saskia Eppinger, and Michael Krautblatter

Chair of Landslide Research, Technical University of Munich, 80333, Germany

*Correspondence to*: Philipp Mamot (Philipp.mamot@tum.de)

**Abstract.** In the last two decades, permafrost degradation has been observed to be a major driver of enhanced rock slope instability and associated hazards in high mountains. While the thermal regime of permafrost degradation in high mountains has already been intensively investigated, the mechanical consequences on rock slope stability have so far not been reproduced in numerical models. Laboratory studies and conceptual models argue that warming and thawing decrease rock and 10 discontinuity strength and promote deformation.

This study presents the first general approach for a temperature-dependent numerical stability model that simulates the mechanical response of a warming and thawing permafrost rock slope. The proposed procedure is applied to a rockslide at the permafrost-affected Zugspitze summit crest. Laboratory tests on frozen and unfrozen rock joint and intact rock properties provide material parameters for the discontinuum model developed with the Universal Distinct Element Code (UDEC). 15 Geophysical and geotechnical field surveys deliver information on the permafrost distribution and fracture network.

The model demonstrates that warming decreases rock slope stability to a critical level, while thawing initiates failure. A sensitivity analysis of the model with a simplified geometry and warming trajectory below 0 °C shows that progressive warming close to the melting point initiates instability above a critical slope angle of 50–62°, depending on the orientation of the fracture network. The increase in displacements intensifies for warming steps closer to zero degree.

The simplified and generalised model can be applied to permafrost rock slopes (i) which warm above -4 °C, (ii), with ice-filled joints, (iii) with fractured limestone or probably most of the rock types relevant for permafrost rock slope failure, (iv) with a wide range of slope angles (30–70°) and orientations of the fracture network (consisting of three joint sets). The presented model is the first one capable of assessing the future destabilisation of degrading permafrost rock slopes.

## 1 Introduction

Rock slope failures in high mountain areas endanger human lives, settlements and infrastructure. The stability of alpine rock slopes can be considerably affected by the climate-induced degradation of bedrock permafrost (Fischer et al., 2006; Gruber et al., 2004; Gruber and Haeberli, 2007; Ravanel and Deline, 2015). In particular, the well-monitored rock slope failures in the Mont Blanc Massif have been linked to the degradation of bedrock permafrost and ice-filled joints (Ravanel et al., 2010; Ravanel and Deline, 2008; Ravanel and Deline, 2011).



Mountain permafrost warmed globally by 0.19 °C between 2007 and 2016 (Biskaborn et al., 2019). Simulations of long-term permafrost evolution in Swiss, German or Norwegian mountain ranges show an overall warming of permafrost and thaw layer deepening up to the end of the century (Gallemann et al., 2017; Hipp et al., 2012; Marmy et al., 2016). Rock slope failures influenced by permafrost degradation are expected to become more frequent as a result of the warming climate (Gobiet et al., 2014; Huggel et al., 2012).

Failure tends to occur preferentially along discontinuities, as they are planes of weakness in the much stronger, intact rock (Wyllie, 2018). Fractures and fractured zones in mountain bedrock permafrost usually contain ice to depths of several tens of metres (Deline et al., 2015; Gruber and Haeberli, 2007). The majority of failure events in permafrost-affected rock exposed ice-filled joints as potential shear and detachment planes at different volumetric scales all over the world: for instance, the 2003 Matterhorn block fall (0.002 Mio m³; Weber et al., 2017), the 2014 Piz Kesch rock slope failure (0.15 Mio m³; Phillips

et al., 2017) and the 2017 Pizzo Cengalo failure with 8 fatalities (3–4 Mio m³; Walter et al., 2020) in Switzerland, the 1987 Val Pola debris avalanche in the Italian Alps (33 Mio m³; Dramis et al., 1995), the 2005 Mt. Steller rock-ice avalanche in Alaska (40–60 Mio m³; Huggel et al., 2010) and the 2002 Kolka/Karmadon rock-ice avalanche with 140 fatalities in the Russian Caucasus (100 Mio m³; Huggel et al., 2005).

The warming of permafrost in rock slopes can reduce the shear resistance along rock joints by (i) reducing the fracture

toughness of cohesive rock bridges, (ii) by lowering the friction along rock-rock contacts, (iii) by altering the creep of ice infillings and (iv) by reducing the fracture toughness of ice fillings and of rock-ice contacts (Krautblatter et al., 2013). Ice fillings contribute to higher rock joint shear strengths in terms of interlocking and adhesion along the rock-ice interface and can increase the stability of a rock slope. This effect is reduced, or even lost, when the temperature approaches the melting point (Davies et al., 2001; Gruber and Haeberli, 2007; Mamot et al., 2018). So far, a huge number of laboratory test series have

been performed on the influence of warming or thawing on rock-ice-mechanical properties to better estimate the consequent effect on rock slope stability: the ductile temperature- and stress-dependent creep of ice and ice-rich soils have been investigated by Arenson and Springman (2005) and Sanderson (1988). Whilst the brittle failure and creep of ice-filled rock joints have been observed by Davies et al. (2000), Günzel (2008) and Mamot et al. (2018), the mechanics of frozen and unfrozen intact rock have been studied by Dwivedi et al. (2000), Inada and Yokota (1984), Kodama et al. (2013) and Mellor

(1973). The friction along frozen and unfrozen rock joints was measured by Krautblatter et al. (2013). A temperature-dependent strength reduction was able to be demonstrated for all these investigated mechanical parameters (Table 1).



**Table 1: A systematic survey of previous laboratory studies on the thawing-dependent mechanical strength reduction of rock and rock joint parameters.**

| Mechanical parameter | Decrease due to warming % °C⁻¹ | absolute | Temperature range tested [°C] | Type of rock / Normal stress | Reference |
|---|---|---|---|---|---|
| **Uniaxial compressive strength [MPa]** | | | | | |
| rock | 1.5 | 200 to 170 | -10 to 0 | granite | Mellor (1973) |
| | 4.7 | 95 to 50 | | sandstone | |
| | 5 | 70 to 35 | | limestone | |
| | 2 | 45 to 30 | -20 to -5 | tuff and andesite | Kodama et al. (2013) |
| | 0.6 | 220 to 165 | -40 to 0 | granite | Inada and Yokota (1984) |
| | 0.9 | 200 to 130 | -40 to 0 | andesite | |
| polycrystalline ice | 1.5 | 6 to 1.5 | -50 to 0 | -- | Inada and Yokota (1984) |
| | 1.7 | 17 to 3 | -50 to 0 | -- | Schulson and Duval (2009) |
| | 5 | 4 to 2 | -10 to 0 | -- | Butkovitch (1954) |
| **Uniaxial tensile strength [MPa]** | | | | | |
| rock | 1.7 | 12 to 10 | -10 to 0 | granite | Mellor (1973) |
| | 6.7 | 12 to 4 | | sandstone | |
| | 6.7 | 12 to 4 | | limestone | |
| | 4 | 5.5 to 2.5 | -20 to -5 | tuff and andesite | Kodama et al. (2013) |
| | 1.1 | 14 to 10 | -20 to 0 | granodiorite | Glamheden (2001) |
| | 0.6 | 12 to 9 | -40 to 0 | granite | Inada and Yokota (1984) |
| | 1.1 | 20 to 11.5 | -40 to 0 | andesite | |
| polycrystalline ice | 0.9 | 2.5 to 1.4 | -50 to 0 | -- | Inada and Yokota (1984) |
| | 1.3 | 1.5 to 1.3 | -10 to 0 | -- | Butkovitch (1954) |
| **Young's modulus [GPa]** | | | | | |
| rock | 0.5 | 42 to 39 | -15 to 0 | granodiorite | Glamheden (2001) |
| **Poisson's ratio** | | | | | |
| rock | 0.7 | 0.27 to 0.24 | -15 to 0 | granodiorite | Glamheden (2001) |
| **Fracture toughness [MPa m⁻⁰·⁵]** | | | | | |
| rock bridges | 0.8 | 1.2 to 0.8 | -10 to 30 | limestone | Dwivedi et al. (2000) |
| | 0.9 | 0.4 to 0.2 | | sandstone | |
| | 0.02 | 1.5 to 1.5 | | basalt | |
| | 0.2 | 1.6 to 1.4 | | dolerite | |
| polycrystalline ice | 0.6 | 0.1 to 0.08 | -50 to -2 | -- | Schulson and Duval (2009) |
| **P-wave velocity [km s⁻¹][a]** | | | | | |
| rock | 2.8 | 6.4 to 3.7 | -15 to 0 | limestone[b] | Draebing and Krautblatter (2012) |
| | 0.8 | 6.1 to 5.4 | | gneiss[b] | |
| | 1 | 6.1 to 5.2 | | granodiorit[b] | |





| Joint shear strength [MPa] | | | | | |
|---|---|---|---|---|---|
| concrete-ice | 18.1 | 0.4 to 0.1 | -5 to -0.5 | 100 kPa | Davies et al. (2000) |
| | 16 | 0.8 to 0.2 | | 1400 kPa | |
| | 31 | 0.5 to 0.2 | -2.5 to 0 | 140 kPa | Krautblatter et al. (2013) |
| | 24 | 0.6 to 0.3 | | 500 kPa | |
| ice-steel-ice | 10.2 | 1.1 to 0.2 | -10 to -2 | -- | Jellinek (1959) |
| rock-ice-rock | 7.4 | 1.0 to 0.3 | -10 to -0.5 | limestone, 100 kPa | Mamot et al. (2018) |
| | 8.2 | 1.4 to 0.3 | | limestone, 200 kPa | |
| | 6.7 | 1.4 to 0.5 | | limestone, 400 kPa | |
| | 17.2 | 1.0 to 0.4 | -4 to -0.5 | limestone, 800 kPa | |
| Joint friction angle [°] | | | | | |
| rock-rock | 0.4 | 43.1 to 38.7 | -4.5 to 22 | dolomite | Krautblatter et al. (2013) |
| | 0.6 | 35.5 to 30 | | limestone | |
| rock-ice-rock | 10 | 64.5 to 27.7 | -8 to -0.5 | limestone, 100-400 kPa | Mamot et al. (2018) |
| | 17 | 30.5 to 13.5 | -4 to -0.5 | limestone, 100-800 kPa | |
| | 4 | 47.7 to 13.5 | -18.5 to -0.5 | granite | Barnes et al. (1971) |
| Joint cohesion [MPa] | | | | | |
| rock-ice-rock | 11.5 | 0.6 to 0.1 | -8 to -0.5 | limestone, 100-400 kPa | Mamot et al. (2018) |
| | 21.2 | 0.7 to 0.2 | -4 to -0.5 | limestone, 100-800 kPa | |

**Note: [a]The P-wave velocity of rock correlates highly to Mode I fracture toughness which correlates closely to Mode II fracture toughness (Chang et al., 2002). [b]P-wave velocities are parallel to cleavage/bedding.**

The necessity to better link permafrost degradation and rock slope stability has been stated several times (Etzelmüller, 2013; Harris et al., 2009; Krautblatter et al., 2012). To our knowledge, the attempt to study the effect of warming or thawing on the
mechanical response of a rock slope has been realised only once by Davies et al. (2001). For this, the authors simulated the warming of a self-constructed laboratory centrifuge model representing a simplified, full rock slope. Factors of safety were estimated for unfrozen and ice-filled joints at temperatures close to the melting point. Included temperature-dependent mechanical properties were derived from laboratory shear tests by Davies et al. (2000).

By now, numerical modelling is a common and established method to (i) mechanically assess rock slope stability and
characterise failure, deformation and influencing factors (Stead et al., 2006), and (ii) thermally analyse the spatial distribution and evolution of permafrost in a mountain or rock slope (Haberkorn et al., 2017; Moore et al., 2011; Myhra et al., 2017; Noetzli and Gruber, 2009). However, no thermo-mechanically combined numerical model has been developed yet which considers the warming- and thawing-dependent deformation and strength reduction of permafrost bedrock and ice-filled discontinuities. Nevertheless, to anticipate failure in a warming climate, we need to understand how rock-ice mechanical components control
rock slope destabilisation and how failure close to melting, and at thawing, can be mechanically expressed in models. Discontinuum discrete-element codes simulate the deformation behaviour of fractured rock masses and consider the influence of jointing (Stead et al., 2006). Among these codes, UDEC (Universal Distinct Element Code) used for distinct-element





modelling of jointed and blocky material in 2D, has so far been applied to a number of stability analyses for unfrozen rock slopes (Bhasin and Kaynia, 2004; Fischer et al., 2010; Gischig et al., 2011a, 2011b, 2011c; Glamheden and Lindblom, 2002; Kveldsvik et al., 2008; Welkner et al., 2010).

In this paper, we present the first numerical discontinuum model that calculates the influence of warming and thawing of frozen rock and rock joints on the mechanical strength and deformation behaviour of an entire rock slope. In this context, we address the following research questions:

1) How can mechanical and thermal data from the field and from the laboratory be combined to develop a coupled numerical stability model for a warming and thawing permafrost rock slope?

2) Is the numerical stability model of the Zugspitze summit crest capable of simulating the warming-dependent changes in stability observed at the laboratory scale?

3) How is the stability of a warming permafrost rock slope dependent on the slope angle and the orientation of the fracture network?

A slowly moving rockslide at the Zugspitze summit crest (2900 m a.s.l.), Germany, with presumably degrading permafrost bedrock was chosen as a real-world example for rock slope instability (Mamot et al., 2018). Rock specimens were collected close by the rockslide and used for a large number of mechanical laboratory tests on a broad spectrum of frozen and unfrozen intact rock and rock joint properties. Intact rock properties were converted to rock mass characteristics using internationally standardised mathematical equations. Electrical resistivity tomography (ERT) was applied to identify current frozen and unfrozen slope sections. These were implemented in the numerical model to spatially assign frozen or unfrozen rock mass and joint parameters to the corresponding model sections and to derive a rough spatial warming pattern.

Considering the numerous laboratory-based observations of the strength reduction of rock-ice-mechanical properties at warming/thawing (Table 1), we expect to model a strong effect on the slope stability. For this, we set up a discontinuum model in 2D with UDEC 7. In a first step, we modelled the numerical impact of warming and thawing on the stability of the Zugspitze summit crest with ice-filled joints. In a second step, we applied the same model to a frozen rock slope with simplified geometry to study the numerical impact of rising subzero temperature on the rock slope stability for varying slope angles and orientations of the joint set configuration.

## 2    A general thermo-mechanically coupled stability model

We propose the following general procedure to develop a numerical discontinuum model to calculate the influence of warming and thawing on the mechanical strength and deformation behaviour of a (partially) frozen rock slope (Fig. 1):



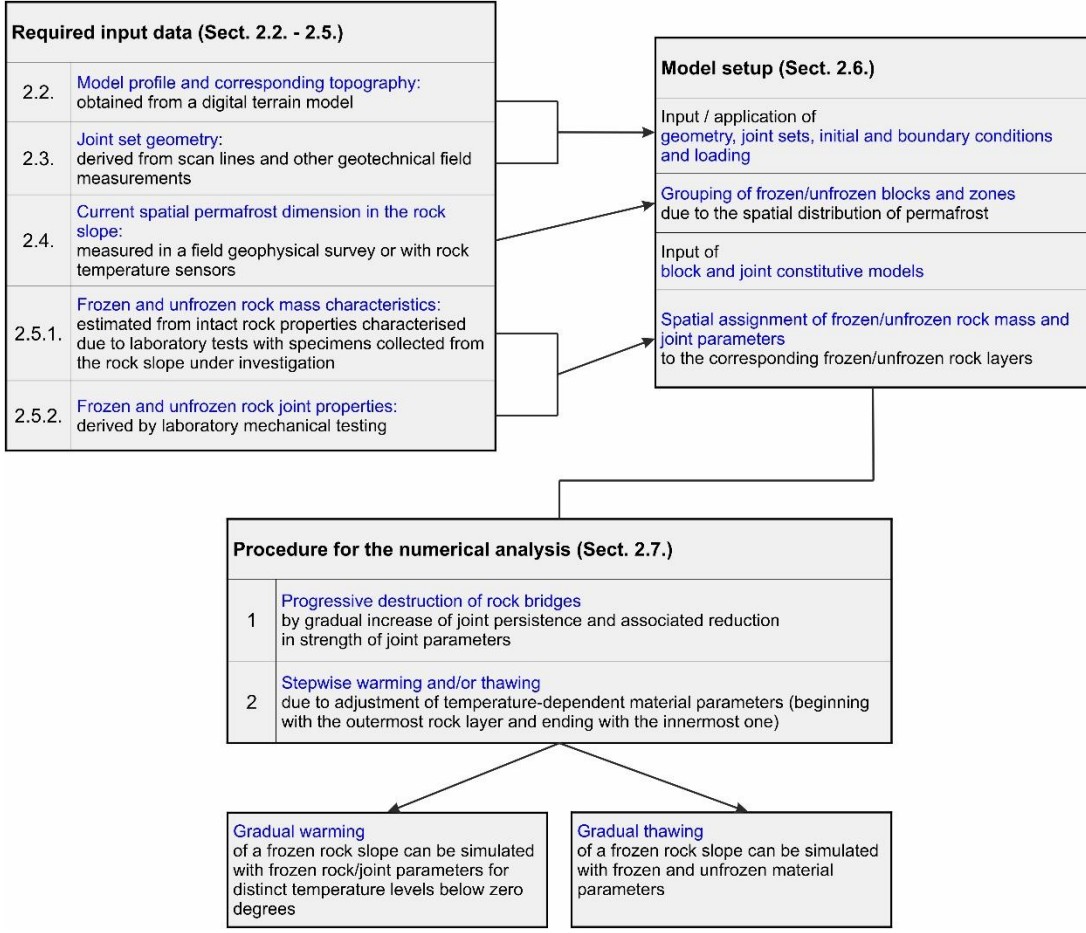

**Figure 1: General procedure for a thermo-mechanically combined discontinuum model for the stability of a warming or thawing permafrost rock slope. The procedure is embedded as methodological approach in this manuscript and, hence, is linked to specific sections.**


## 2.1 Application to the Zugspitze summit crest

The presented modelling approach (Fig. 1) was applied to a real-world example of alpine rock slope instability potentially affected by permafrost degradation: The topography, model profile, joint set geometry, rock samples and permafrost distribution for the numerical model were taken from a shallow rockslide at the south-face of the Zugspitze summit crest

(2900 m a.s.l., Fig. 2) in Germany. The unstable rock mass was assessed to a volume of approximately $2.9 * 10^4$ m³ (Fig. S2). Measured fracture displacements between 2013 and 2019 show that the rock mass creeps slowly at an average of 2.1 mm yr$^{-1}$, with deformation rates reducing by 84 % when comparing summer months (June to September) with the remaining seasons (Fig. S5). Details on the volume estimation and the measurement of the crack displacements are given in the supporting

material.



The Zugspitze is one of the highest peaks of the Northern Calcareous Alps, located in the Eastern Alps. The summit crest consists of Triassic Wetterstein limestone which has a thickness of 1000 m and constitutes the majority of the Zugspitze massif and the Wetterstein mountains (Gallemann et al., 2017; Miller, 1962). The limestone is massy, fine-grained and dolomised and shows little heterogeneity in terms of lithological properties (Krautblatter et al., 2010). Its porosity ranges between 1.9 %
(Draebing and Krautblatter, 2012) and 4.4 % (Krautblatter et al., 2010).

## 2.2 Model profile

The numerical model was set up in 2D and, thus, required a cross section of the summit crest. This profile covers a distance of 100 m, it runs from the north-face, across the crestline, to the south-face, and it crosses one of the main shear zones of the rockslide (Fig. 2).




Earth **Surface**
**Dynamics**
Discussions

Figure 2: (a) View on the south-face of the Zugspitze summit crest, Germany. The study site is located 60 m below the peak (2960 m a.s.l.), highlighted by the red box. (b) View from above down on the study site, including the assumed main shear zones of the upper part of the rockslide (red dashed lines), the profile used for the numerical model (black line) and the transect for the electrical resistivity survey (black dashed line). (c) Location of the Zugspitze in the context of the European Alps. The digital elevation model was developed with SRTM data from CGIAR CSI and has a cell size of 90 m (Jarvis et al., 2008).

## 2.3 Rock joint geometry and kinematic analysis

Four discontinuity sets (K1, K2, K3, K4) were identified due to scan lines and field mapping (Fig. 3a). The trend of the profile for the UDEC model was chosen to strike at 146°, which is more or less perpendicular to the estimated orientation of the southern slope face (45/160). Hence, K2 (33/063) had to be excluded from the numerical analysis as the dip direction deviated by 66.5° from the trend of the model profile. The remaining joint sets deviated by 20–33° and were implemented in the model





as the standard deviations of their dip directions ranged between 4–20°, falling within the tolerable range of 30° deviation (Table 2). Joint set K1 represents the bedding planes and daylights in the south-face at an angle of 24° (Fig. 3b).


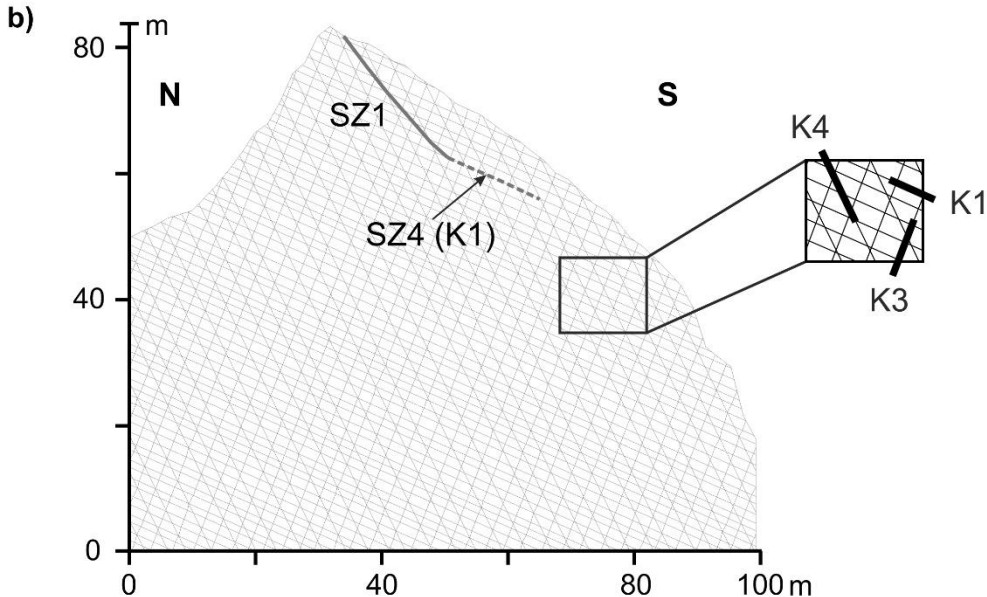

Figure 3: (a) Orientations of field-measured discontinuities in a stereographic projection, calculated with DIPS 7.0 (Rocscience). The red great circles represent the joint sets that were selected as input for the numerical model. The contours depict the Fisher concentrations (density) of the poles. K = Joint set. (b) Fracture network, extent and topography of the 2D numerical model. Shear
zones (SZ) included in the model are marked with thick grey lines. The dashed line represents the uncertain course of the potential failure plane at lower slope sections.



**Table 2: Mapped geometric discontinuity characteristics. Standard deviations are only given for measured parameters. K1-4 = Joint sets. SZ = Shear zone.**

| Geometric parameter | K1 | K3 | K4 | SZ1 (K4) | Input for numerical model |
|---|---|---|---|---|---|
| Dip/dip direction [°] | 24/175 | 69/293 | 66/126 | 63/130 | x |
| Deviation of dip direction from trend of model profile [°] | 29.3 | 33.3 | 20.4 | 16.0 | x (as below 30°, considering the standard deviation) |
| Standard deviation of dip direction [°] | 20.2 | 12.6 | 4.5 | -- | x |
| Standard deviation of dip [°] | 2.9 | 3.8 | 7.5 | -- | |
| Spacing [m] | 0.27 ± 0.3 | 0.54 ± 0.6 | 0.59 ± 0.7 | -- | x |
| Aperture [mm] | 7.0 ± 17.1 | 2.9 ± 6.8 | 5.3 ± 8.1 | 270.0 | |
| Joint frequency [F m$^{-1}$] | 3.7 | 1.9 | 1.6 | -- | x (simplified representation) |
| Joint roughness coefficient JRC[a] | 5.5 ± 1.8 | 8.6 | -- | 4.4 ± 3.3 | |
| Number of mapped joints | 15 | 37 | 19 | 1 | |

**Note: [a]The JRC was measured along joints not included in the scanlines.**

UAV-based (Unmanned Aerial Vehicle) photogrammetry was performed at the Zugspitze summit crest to compute a 3D point cloud providing information on the possible failure mechanism and shear zones which delimit the currently observed unstable rock mass. In addition, a simple kinematic analysis of a potential plane and wedge failure was conducted with DIPS 7.0

(Rocscience) involving the southern slope-face and the main shear zones, which were identified due to field mapping and the preceding analysis of the point cloud. A detailed description of both methods is given in the supplement material. The results showed that the most likely failure mechanism is a complex combination of a plane and a wedge failure (Fig. S2, Fig. S3). In the upper part of the slope instability, wedge failure occurs along shear zones SZ1/SZ3 including a tension crack SZ2, while local planar sliding along shear zone SZ1 may support the displacement (Fig. 2b). At lower slope sections, two further shear

zones (SZ4 and SZ5) were determined to form the assumed downslope boundary of the unstable rock mass (Fig. S2). Here, wedge failure may occur along shear zone SZ3 and a stepped plane constituted of shear zones SZ4 and SZ5, while planar sliding along SZ4 or SZ5 potentially enhances the failure process.

To allow simplification and a representation in 2D, shear zone SZ1 (equivalent to K4) and shear zone SZ4 (corresponds to K1) were chosen to be included in the numerical model (Fig. 3b). We assume the slope instability to be significantly dominated by

shear zone SZ1. Observed maximum displacements along the upper part of SZ1 lie in the range of decimetres and point to a high level of joint persistence. Further, three carst dolines with a depth of several metres were mapped along SZ1 and at the intersection of shear zones SZ1 with SZ2 (Fig. S4). SZ1 dips to the southeast and corresponds to joint set K4. According to geotechnical field mapping, it has a trace length of approximately 70 m and runs in a maximum depth of 10–15 m. It is highly





fractured and at some places it opens to a width of several decimetres, filled with fine material which ranges from clay- to gravel-size.

## 2.4 Spatial permafrost distribution within the summit crest

In the 1960s and 1990s, the occurrence of permafrost at the Zugspitze summit was able to be confirmed by deep and permanently ice-filled fractures with a maximum aperture of 0.1 m (Körner and Ulrich, 1965; Ulrich and King, 1993). The Zugspitze summit area is located at the lower permafrost extension limit. Current borehole temperatures at the peak of the Zugspitze average -1.3 °C within the permafrost core area (20–25 m away from the rock walls). At the margin to the active layer (ca. 5 m away from the north-face), the temperatures approach minima of -6 °C during winter (Böckli et al., 2011; Gallemann et al., 2017; Krautblatter et al., 2010; Noetzli et al., 2010).

Laboratory-calibrated electrical resistivity tomography (ERT) is capable of characterising and monitoring the spatial variability and evolution of mountain bedrock permafrost in steep rock walls (Keuschnig et al., 2017; Krautblatter et al., 2010; Murton et al., 2016). The outlined procedure of the measuring setup in the field was similarly applied to the Zugspitze summit crest to accurately assess the distribution of permafrost along the model profile (Fig. 2b). Two surveys were conducted at the end of August 2014 and 2015, when the active layer reached the maximum thickness (Gallemann et al., 2017). Data processing and inversions were performed with Res2Dinv. To assign the measured ER of the summit crest to frozen or unfrozen slope sections, we used the characteristic laboratory ER of saturated frozen and unfrozen Wetterstein limestone, defined (i) by Krautblatter et al. (2010) based on one rock sample, and (ii) in the context of this study based on two rock blocks from the field site. For the laboratory tests, we followed the suggested procedure by Krautblatter et al. (2010), though limited to a single cooling and subsequent freezing trajectory of the samples from 10 down to -6 °C. The supplement material provides additional information on the laboratory tests as well as on the measuring setup, data acquisition and analysis of the field surveys.

The laboratory ER clearly showed values lower than 19 kΩm for unfrozen rock and values higher than 28–29 kΩm for frozen rock (Fig. S6). The equilibrium freezing point at -0.2 or -0.4 °C is indicated by a 21- or a 37-fold increase of the frozen temperature-resistivity gradient. This pattern of electrical resistivity versus rock temperature is similar to the one demonstrated by Krautblatter et al. (2010): here, the ER measures 30 ± 3 kΩm at the equilibrium freezing point (-0.5 °C). Overall, we defined ER-values lower than 19 kΩm and higher than 33 kΩm to be unfrozen or frozen, respectively. The range of values lain inbetween was defined to be possibly frozen rock.



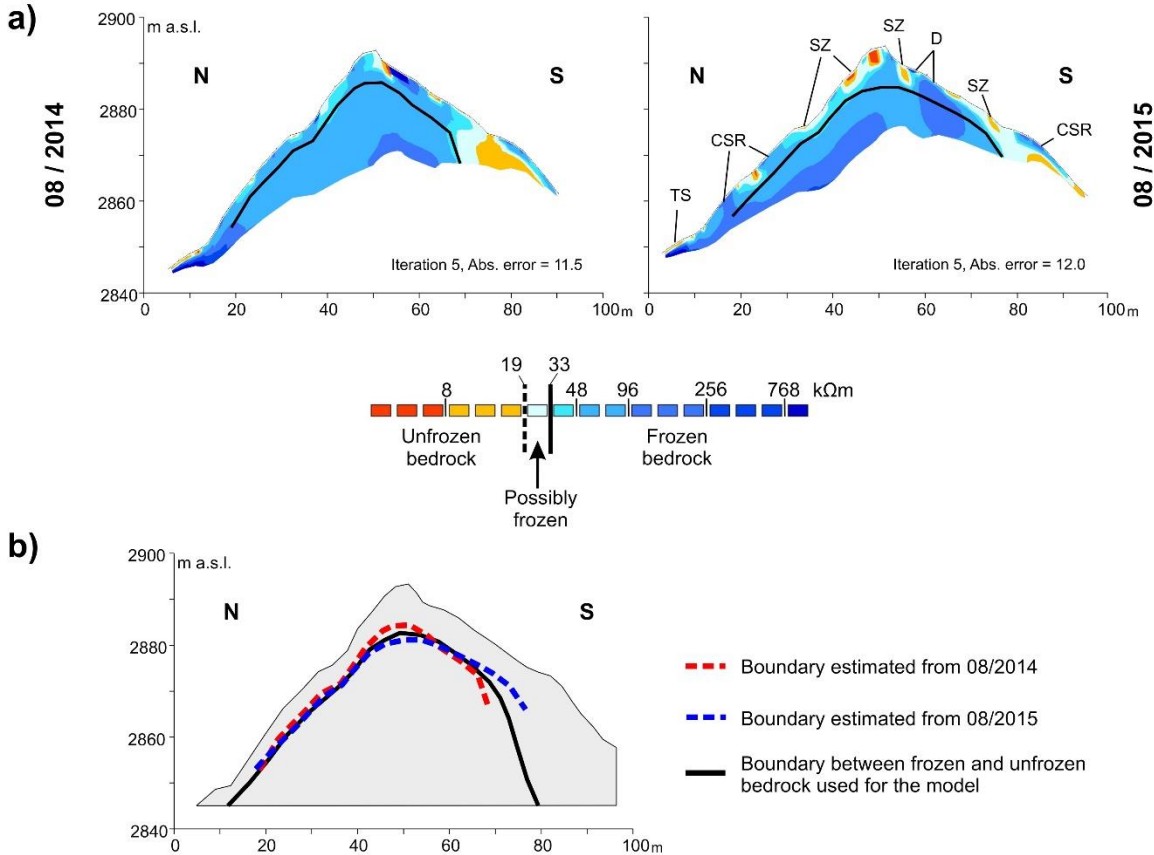

**Figure 4: (a) Distribution of frozen and unfrozen bedrock in the Zugspitze summit crest derived from two ER surveys in August 2014 and August 2015. TS = Talus slope. CSR = Compact solid unfrozen rock. SZ = Shear zone. D = Doline. The locations of these features are the same in the tomography of 2014. (b) Combination of the ERT campaigns for the estimation of the current spatial extent of bedrock permafrost implemented in the numerical model (Fig. 5).**


The ER inversions of 2014 and 2015 are interpreted as follows (Fig. 4a): The shallow rock layer is almost continuously unfrozen on both aspects of the crest. Interstitial patches of resistivities higher than 33 kΩm do not represent frozen rock; instead, they point to massive ice measured in the deep carst doline close to the crestline, or to sections of compact solid rock

or to bad electrode-ground coupling (Hauck & Kneisel, 2008). The assumption of the unfrozen near-surface rock layer is confirmed by thermistor-based rock temperatures measured in a depth of 10–80 cm below the ground surface at the same time as the ERT surveys. The corresponding thermistors (iButtons) were installed along the ERT transect and show that the rock temperatures do not remain below zero degrees throughout the year (Fig. S4, Fig. S7). Furthermore, rock temperatures along the borehole at the Zugspitze peak show a 2–4 m thick active layer on the north-face (Gallemann et al., 2017). Due to this

evidence, the current boundary to permafrost on the north-face and at the upper part of the south-face was set to a continuous depth of approximately 5 m (black line in Fig. 4b). On the lower south-slope, the unfrozen rock can be found down to a depth of 10 m and more indicated by resistivities < 19 kΩm. The individual patches of low electrical resistivity (< 19 kΩm) refer to





major shear zones or to the highly porous talus slope below the northern rock wall. When fractures or the debris of the talus
slope are filled with air or with unfrozen, saturated and fine material, the corresponding electrical resistivities are much lower
than for ice fillings (Hauck and Kneisel, 2008; Supper et al., 2014). Unfrozen fine fillings permit water to infiltrate easily and
can represent zones of high conductivity (Supper et al., 2014), particularly in the case of preceding precipitation or snow melt
(Keuschnig et al., 2017). Having defined the boundaries between frozen and unfrozen slope sections for two successive years
2014 and 2015, the resulting overall spatial extent of permafrost relates to the core of the crest and rock layers below 5 m at
the north-face, while the layer of unfrozen bedrock at the south-face becomes thicker with decreasing altitude (Fig. 4b).

## 225 **2.5 Mechanical properties**

The UDEC model software simulates the mechanical behaviour of a discontinuous medium: the assemblage of discrete rock
blocks separated from each other by discontinuities. As such, the model requires material properties for both the rock blocks
and discontinuities (Itasca Consulting Group, 2019). For stability analyses at the scale of a full rock slope, the appropriate rock
strength to be used for model parameter input is that of the rock mass, and not of an intact rock specimen for laboratory tests.
Discontinuities reduce rock mass strength and induce non-linearities and anisotropy in the stress-strain behaviour (Hoek et al.,
2002; Wyllie, 2018). Hence, blocks and zones were allocated rock mass properties to account for the influence of
heterogeneities, micro-fractures and other small discontinuities on the mechanical response of the bigger rock blocks.

To assess the effect of thawing on the mechanical strength and deformability, and to determine the required material parameters
for the model, we tested an extensive number of frozen and unfrozen rock and joint properties of the field site material in the
laboratory. The tests were performed with Wetterstein limestone samples collected at the study site. Frozen rock samples had
a mean temperature of -5 °C, while unfrozen rock specimens were tested at room temperature (~22 °C). As we assume the
rock mass of a real-world rock slope to be usually saturated, the frozen and unfrozen rock samples were mostly tested in a
fully saturated state. Detailed information on the preparation of the rock samples, the test setups and conditions can be looked
up in the supplements.

## 240 **2.5.1 Frozen and unfrozen intact rock and rock mass**

The strength and deformability of the model blocks and zones within the blocks were represented by the Mohr-Coulomb
plasticity constitutive model with a tension cut-off. Assuming the blocks and zones to represent the rock mass, the constitutive
law requires the following input parameters for the rock mass: the mass density $\rho$, the elastic bulk and shear moduli ($K_m$ and
$G_m$), the cohesion $c_m$, the internal angle of friction $\varphi_m$ and the tension limit $\sigma_{tm}$.
The density $\rho$, the Poisson's ratio $\upsilon$ and the dilatational wave $V_D$ were measured in the laboratory for calculating the elastic
moduli of the intact rock $E$ (Young's modulus), $K$ and $G$. These were later used to determine the joint stiffness parameters
(Sect. 2.5.2.). The lab results of the Poisson's ratio, and additional lab tests on the intact rock uniaxial compressive strength $\sigma_c$
were used to quantify elastic moduli and the uniaxial tensile strength $\sigma_{tm}$ of the rock mass (see below).

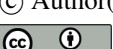



The mean dilatational wave velocity and the Poisson's ratio of the intact rock were derived by ultrasonic tests. Then, the
dynamic Young´s modulus was given by

$$E_{dyn} \, [\text{GPa}] = \rho * V_D^2 \qquad \qquad \text{Eq. (1)}$$

with the rock density $\rho$ (g/cm³) and the dilatational wave velocity $V_D$ (m/s) (Rentsch and Krompholz, 1961). The elastic shear
and bulk moduli for the intact rock were calculated with the following equations (Tipler and Mosca, 2004):

$$G \, [\text{GPa}] = \frac{E_{dyn}}{2 *(1+v)} \qquad \qquad \text{Eq. (2)}$$

and

$$K \, [\text{GPa}] = \frac{E_{dyn}}{3 *(1-2* v)} \qquad \qquad \text{Eq. (3)}$$

Here, the Poisson's ratio $v$ describes the ratio of the transverse strain to the longitudinal strain under conditions of uniaxial
stress (Jaeger et al., 2007).

In the next step, the GSI (Geological Strength Index) rock mass classification system of Hoek and Brown (1997) was used to
transfer the intact rock data to rock mass characteristics and switch to field-scale properties. The GSI system is directly linked
to Mohr-Coulomb or Hoek-Brown strength parameters and rock mass deformation moduli (Cai et al., 2004; Hoek and Brown,
1997). It is semi-quantitative and requires a field survey of the rock mass including description and measurement of block
volumes and joint characteristics. Following this scheme, the deformation modulus of the rock mass was given by

$$E_m \, [\text{GPa}] = \left(1 - \frac{D}{2}\right)\sqrt{\sigma_c}/100 * 10^{((GSI-10)/40)} \qquad \qquad \text{Eq. (4)}$$

where *GSI* is the geological strength index for the rock mass, *D* is the disturbance factor and $\sigma_c$ is the uniaxial compressive
strength of the intact rock (Hoek et al., 2002).

A *GSI* value of 65 was derived from the proposed chart by Cai et al. (2004; Fig. 7) relating to a blocky to very blocky rock
mass formed by four joint sets with a mean spacing of 56 cm, a mean persistence of and rough, slightly weathered surfaces.
The disturbance factor *D* was set to 0 as no significant blasting or excavating actions have been performed in the near
surroundings of the profile.

Values for the rock mass bulk and shear moduli were derived according to Eq. (2) and (3) with the respective $E_m$-values. The
rock mass internal friction angle $\varphi_m$ and cohesion $c_m$ were estimated with the *GSI* and the Hoek-Brown constant $m_i$, following
the strategy by Cai et al. (2004; Fig. 5) which is valid for $\sigma_3 < 5$ MPa. Since the temperature-dependent uniaxial compressive
strength is included in the equations for the cohesion, rock mass cohesion values could be determined for both thermal states.
However, the frozen friction angle was set to the same value as the unfrozen one because the respective relation is not
temperature-dependent. The uniaxial tensile strength $\sigma_{tm}$ was calculated with the following relation

$$\sigma_{tm} = \frac{\sigma_c}{2} \left(m_b - \sqrt{m_b^2 + 4s}\right) \qquad \qquad \text{Eq. (5)}$$





where $\sigma_c$ is the uniaxial compressive strength of the intact rock and $m_b$ and $s$ are constants for the rock mass (Hoek and Brown, 1997).


**Table 3: Laboratory-tested strength reduction of intact dolomised Wetterstein limestone due to thawing. Standard deviations (indicated with ±) are given for measured parameters, and they were used for determination of minimum and maximum values (given in parentheses) of the calculated parameters. RMC refers to parameters that are used for rock mass characterisation.**

| Mechanical parameter | Saturated frozen (-5 °C) | Saturated unfrozen (+22 °C) | Decrease due to thawing % | % °C$^{-1}$ | Test / equation applied | RMC |
|---|---|---|---|---|---|---|
| Density $\rho$ [g/cm³] | -- | $2.7 \pm 0.01$ | -- | -- | Weighing tests in water bath | x |
| Porosity $n$ [%] | -- | $0.9 \pm 0.4$ | -- | -- | Weighing tests in water bath | |
| Young's modulus $E$ [GPa] | 84.3 (81.7/85.3) | 67.1 (55.7/77.6) | 20.4 | 0.8 | Eq. (1), after Tipler and Mosca (2004) | |
| Shear modulus $G$ [GPa] | 32.4 (31.7/32.6) | 25.8 (21.9/29.2) | 20.4 | 0.8 | Eq. (2), after Tipler and Mosca (2004) | |
| Bulk modulus $K$ [GPa] | 70.3 (64.8/74.8) | 55.9 (40.4/76.0) | 20.5 | 0.8 | Eq. (3), after Tipler and Mosca (2004) | |
| Poisson's ratio $\upsilon$ | $0.3 \pm 0.01$ | $0.3 \pm 0.03$ | 0 | 0 | Ultrasonic tests | x |
| Dilatational wave velocity $V_D$ [m/s] | $5560 \pm 50$ | $4950 \pm 400$ | 11.0 | 0.4 | Ultrasonic tests | |
| Uniaxial tensile strength $\sigma_t$ [MPa] | $9.0 \pm 1.4$ | $7.2 \pm 1.9$ | 20.0 | 0.7 | Brazilian tests | |
| Uniaxial compressive strength $\sigma_c$ [MPa] | $109 \pm 25$ | $91 \pm 27$ | 16.5 | 0.6 | Uniaxial compressive strength tests | x |

**Table 4: Estimated and calculated strength reduction of Wetterstein limestone rock mass due to thawing, derived from the GSI scheme after Hoek and Brown (1997). Minimum and maximum values of calculated parameters are given in parentheses. These were determined with the standard deviations of the measured parameters (Table 3). IP = used as input parameter for the numerical model.**

| Mechanical parameter | Saturated frozen (-5 °C) | Saturated unfrozen (+22 °C) | Decrease due to thawing % | % °C$^{-1}$ | Test / equation applied | IP |
|---|---|---|---|---|---|---|
| Shear modulus $G_m$ [GPa] | 9.5 (8.4/10.5) | 8.7 (7.5/9.7) | 8.4 | 0.3 | Eq. (2), after Tipler and Mosca (2004) | x |
| Bulk modulus $K_m$ [GPa] | 20.6 (17.2/24.1) | 18.9 (13.7/25.3) | 8.3 | 0.3 | Eq. (3), after Tipler and Mosca (2004) | x |
| Young's modulus $E_m$ [GPa] | 24.8 (21.7/27.5) | 22.6 (19.0/25.8) | 8.9 | 0.3 | Eq. (4), after Hoek et al. (2002) | |
| Uniaxial tensile strength $\sigma_{tm}$ [MPa] | -0.9 (0.7/-1.1) | -0.7 (-0.5/-0.9) | 22.2 | 0.8 | Eq. (5), after Hoek et al. (2002) | x |
| Friction angle $\varphi_m$ [°] | 44[a] | 44 | 0 | 0 | estimated after Cai et al. (2004) | x |





| | | | | | | |
|---|---|---|---|---|---|---|
| Cohesion $c_m$ [MPa] | 3.9 | 3.3 | 15.4 | 0.6 | estimated after Cai et al. (2004) | x |
| **Parameter of the Hoek-Brown strength criterion** | | | | | | |
| *GSI value* | -- | 65 | -- | -- | estimated after Marinos and Hoek (2000) | |
| $m_i$ | -- | 9 | -- | -- | estimated after Marinos and Hoek (2000) | |
| $m_b$ | -- | 2.6 | -- | -- | calculated after Hoek et al. (2002) | |
| $s$ | -- | 0.02 | -- | -- | calculated after Hoek et al. (2002) | |
| Disturbance factor $D$ | -- | 0 | -- | -- | estimated after Hoek et al. (2002) | |

**Note: [a]The frozen rock mass friction angle was not estimated. It was given the same value as for the unfrozen friction angle, resulting**
**in a decrease in thawing of zero.**

Virtually all of the laboratory-tested or calculated properties of the intact rock and the rock mass decrease between 0.3 and 0.8 % °C$^{-1}$ (8–22 %) upon thawing (Table 3, Table 4). Similar patterns have been observed in mechanical studies of thawing rock summarised in Table 1. As expected, the absolute values are higher for the intact rock. The uniaxial tensile strength and

the elastic moduli were determined for both the intact rock and the rock mass. Here, the tensile strength decreases by a similar amount, while the increase in deformability is more pronounced for the intact rock (0.8 % °C$^{-1}$; 20.4–20.5 %) than for the rock mass (0.3 % °C$^{-1}$; 8.3–8.9 %). This difference can be explained by the small variation in frozen and unfrozen $E_m$-values caused by the similar frozen and unfrozen intact rock uniaxial compressive strength (Eq. (4)).

### 2.5.2 Frozen and unfrozen rock joints

Rock joint parameters were collected for stability analyses with ice-free joints and ice-filled joints. In UDEC, the strength and deformability of rock discontinuities were simulated by the Mohr-Coulomb area contact constitutive model which requires the cohesion $c$, friction angle $\varphi$, joint normal stiffness $k_n$ and joint shear stiffness $k_s$ as input parameters.

The deformability of the joints is described by the joint normal stiffness $k_n$ and joint shear stiffness $k_s$:

$$k_n \text{ [MPa/m]} = \frac{E_m * E}{s * (E - E_m)} \qquad \text{Eq. (6)}$$

where $s$ is the joint spacing (m) and $E_m$ and $E$ are the rock mass, and intact rock Young's moduli (GPa) respectively (Barton, 1972). Since the ratio of $E/G$ is analogous to the $k_n/k_s$ ratio, the joint shear stiffness is given by

$$k_s \text{ [MPa/m]} = \frac{G_m * G}{s * (G - G_m)} \qquad \text{Eq. (7)}$$

where $G_m$ and $G$ are the rock mass and intact rock shear moduli (GPa) respectively (Kulatilake et al., 1992).

To determine the cohesion $c$ and friction $\varphi$ of frozen ice-filled rock joints, we utilised the brittle failure criterion presented by
Mamot et al. (2018):





$$c\,[kPa] = 104.5 - 143.8 * T \qquad\qquad \text{Eq. (8)}$$

and

$$\tan\,[\varphi] = 0.19 - 0.1 * T \qquad\qquad \text{Eq. (9)}$$

where $T$ is the temperature of the rock at failure (Mamot et al., 2018). Equations (8) and (9) are valid for temperatures ranging

from -4 to -0.5 °C and normal stresses between 100 and 800 kPa. As such, the cohesion and friction angle were calculated for temperatures -4, -3, -2, -1 and -0.5 °C, which are currently measured in the frozen sections of the Zugspitze summit (Gallemann et al., 2017). The stress range of 100–800 kPa resembles the rock overburden represented by the model of the Zugspitze summit crest (Fig. 3b).

Values for the cohesion of frozen and unfrozen, ice-free rock joints were roughly estimated from (Krautblatter et al., 2013).

The frictional strength of rock-rock contacts was represented by the residual friction angle $\varphi_r$, as initial displacements lead to the destruction of asperities and smoothing of joint surfaces. Weathering of the surfaces sets in with the progressive destruction of rock bridges and the successive contact and reaction with infiltrating water from precipitation or snow melt. The residual friction angle $\varphi_r$ was estimated with the suggested equation by Barton & Choubey (1977):

$$\varphi_r = (\varphi_b - 20°) + 20 * \left(\frac{r}{R}\right) \qquad\qquad \text{Eq. (10)}$$

where $\varphi_b$ is the basic friction angle, $r$ is the Schmidt hammer rebound value of weathered surfaces, and $R$ is the Schmidt hammer rebound value of unweathered, sawn surfaces. To determine the basic friction angle, tilt tests of frozen and unfrozen joint surfaces were conducted in this study following the recommendations by Barton and Choubey (1977) and Barton (2013). Rebound hardness values of weathered frozen and unfrozen Wetterstein limestone surfaces were collected with the Schmidt hammer (N-type) following the proposed method by Aydin et al. (2005) and Ulusay (2015).



Earth **Surface** Dynamics Discussions — Open Access — EGU


**Table 5: Laboratory-tested and calculated strength reduction of saturated Wetterstein limestone discontinuities due to warming or thawing. Standard deviations (indicated with ±) are given for measured parameters, and they were used for determination of minimum and maximum values (given in parentheses) of the calculated parameters. IP = used as input parameter for the model.**

| Joint mechanical parameter | Type of joint filling | Sub-zero temperature [°C] | Frozen — Joint set / zone — K1 | K3 | K4 / shear | Unfrozen (+22 °C) — zone — K1 | K3 | K4 / shear | Decrease due to thawing/warming % | %°C⁻¹ | Reference / Equation applied | IP |
|---|---|---|---|---|---|---|---|---|---|---|---|---|
| Normal stiffness [MPa m⁻¹] | ice-free/ice-filled | -5 | 26600 (21900/30000) | 11100 (9400/12900) | 13000 (11000/15000) | 25300 (23300/28600) | 10800 (9100/12300) | 12600 (10700/14300) | 2.8 | 0.1 | Eq. (6), calculated after Barton (1972) | x |
| Shear stiffness [MPa m⁻¹] | ice-free/ice-filled | -5 | 10000 (8500/11400) | 4300 (3600/4900) | 5000 (4300/5700) | 9700 (8400/10700) | 4200 (3600/4600) | 4900 (4200/5400) | 2.6 | 0.1 | Eq. (7), calculated after Kulatilake et al. (1992) | x |
| Cohesion [MPa] | ice-filled | -4 | 0.68 (0.45/0.91) | — | — | — | — | — | 74.1 | 21.2. | Eq. (8), calculated after Krautblatter et al. (2013) | x |
|  |  | -3 | 0.54 (0.35/0.72) |  |  |  |  |  |  |  |  |  |
|  |  | -2 | 0.41 (0.25/0.54) |  |  |  |  |  |  |  |  |  |
|  |  | -1 | 0.26 (0.15/0.35) |  |  |  |  |  |  |  |  |  |
|  |  | -0.5 | 0.18 (0.10/0.26) |  |  |  |  |  |  |  |  |  |
|  | ice-free | -4.5 |  | 0.00 |  |  | 0.00 |  | 0 | 0 | estimated after Krautblatter et al. (2013) | x |
| Peak friction angle [°] | ice-filled | -4 | 30.5 (26.1/34.6) | — | — | — | — | — | 59.3 | 17.0 | Eq. (9), calculated after Mamot et al. (2018) | x |
|  |  | -3 | 26.4 (22.3/29.7) |  |  |  |  |  |  |  |  |  |
|  |  | -2 | 22.0 (18.3/24.2) |  |  |  |  |  |  |  |  |  |
|  |  | -1 | 16.8 (14.0/18.3) |  |  |  |  |  |  |  |  |  |
|  |  | -0.5 | 13.5 (11.9/15.1) |  |  |  |  |  |  |  |  |  |
| Rebound value for dry unweathered, sawn surfaces | ice-free | -5 |  | 55.0 ± 1.0 |  |  | 51.9 ± 1.0 |  | 1.6 | 0.1 | Schmidt Hammer tests |  |
| Rebound value for wet weathered surfaces | ice-free | -5 |  | 53.0 ± 2.1 |  |  | 52.1 ± 0.8 |  | 1.7 | 0.1 | Schmidt Hammer tests |  |
| Basic friction angle [°] | ice-free | -5 |  | 38.3 ± 4.4 |  |  | 30.0 ± 4.7 |  | 21.7 | 0.8 | Tilt tests |  |



| Residual friction angle [°] | ice-free | | | | | Eq. (10), calculated after Barton and Choubey (1977) | x |
|---|---|---|---|---|---|---|---|
| | -4.5 | 37.6 (32.8/42.4) | 30.1 (25.5/34.7) | 22.1 | 0.8 | | |





The investigated and calculated joint deformation and mechanical strength properties reduced by 2 to 22 % upon thawing
(corresponding to a decrease by 0.1–0.8 % °C$^{-1}$; Table 5). The small variation in frozen and unfrozen $E_m$-values was also the
reason for the small differences in the frozen and unfrozen joint normal and shear stiffness (0.1 % °C$^{-1}$; 2.6–2.8 %). However,
the stiffness values vary more strongly referring to the various joint sets which are additionally dependent on the specific joint
spacing. In general, the measured unfrozen joint normal and shear stiffness lie well within the range of values proposed by the
UDEC database (Itasca Consulting Group, 2019) and Kulatilake et al. (1992), or measured by Barton (1972) and Bandis et al.
(1983). Similarly, the measured unfrozen joint basic friction angle corresponds well to the values listed e.g. in Barton and
Choubey (1977).

## 2.6 Model setup

The topography of the rock slope was derived from a digital terrain model of the Zugspitze summit area. Joint sets K1, K3,
K4 and the dominant shear zone were included (Fig. 3b). For simplification and shorter computation time, the joint spacing
was chosen to be five-times greater than in reality. All joint sets were created as fully persistent. The presence of intact rock
bridges was accounted for according to the approach by Jennings (1970). After this, the joint cohesion and friction angle were
proportionately increased by the same values of the rock mass dependent on the estimated percentages of rock bridges and
joints within the rock slope. However, the joint stiffness values were not changed for the different degrees of joint persistence.
The blocks in the model are supposed to be deformable and were subdivided into a mesh of finite-difference elements. The
size of the mesh was determined to be 1.5 m. The stress field was initialised according to the varying density of overburden
which depends on the topography of the slopes. We assumed a horizontal to vertical stress ratio of 0.5. Roller boundary
conditions were implemented at the sides of the model (i.e. vertical movements were allowed), whereas vertical and horizontal
displacements were suppressed along the base.
The combined boundary of bedrock permafrost within the summit ridge, derived from ER tomographies of 2014 and 2015
(Fig. 4b), was used to define the current frozen and unfrozen sections of the numerical model. Then, we introduced six
subsurface layers to simulate a stepwise warming or thawing from the slope surface to the core of the crest by adjusting the
temperature-dependent material parameters for a specific temperature level and subsequent numeric cycling. The layers are
oriented parallel to the derived permafrost boundary and account for a stronger warming signal directed from the south-slope
(Fig. 5). The defined current permafrost boundary and the estimated spatial pattern of layers for warming are both in accordance
with modelled current and future thermal fields of arbitrary mountain ridge geometries (Noetzli et al., 2007) and, in particular,
of the Zugspitze (Böckli et al., 2011; Noetzli, 2008).





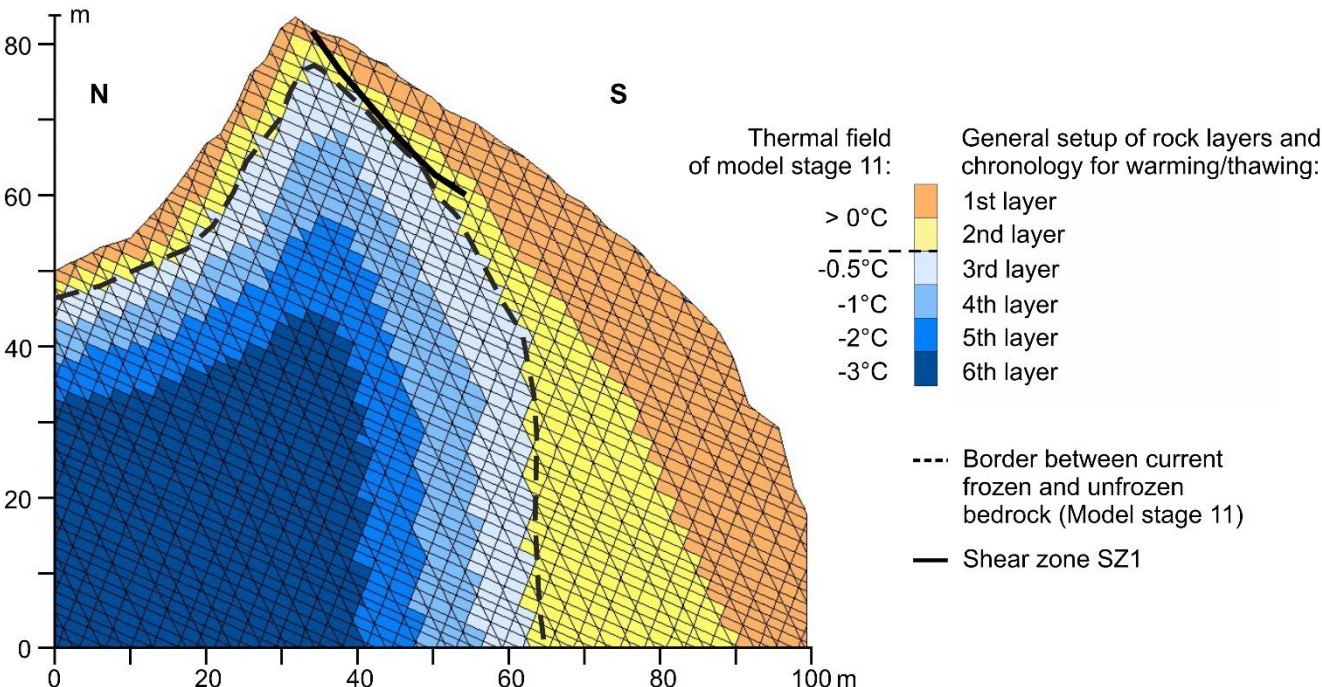

**Figure 5: Spatial warming/thawing pattern for the numerical model of the Zugspitze summit crest. For each warming/thawing step, the six rock layers (coloured areas) are assigned an individual temperature with a characteristic set of mechanical parameters. Warming or thawing proceeds from the outside to the inside of the crest. The dashed black line marks the current boundary between frozen and unfrozen rock, derived in Section 2.4. The presented thermal field is estimated and refers to model stage 11 in Table 6 (current state of permafrost distribution in the crest).**

## 2.7 Procedure for the numerical analysis

The general modelling procedure consists of the calculation of an initial equilibrium and three successive steps of potential destabilisation – the progressive destruction of intact rock bridges, the warming and the thawing of a permafrost rock slope:

**Table 6: Modelling strategy for the Zugspitze summit crest. The model was run with varying rock mass temperature, joint persistency and filling. The temperature levels for gradual warming were: -4, -3, -2, -1 and -0.5 °C.**

| Model stage | Thermal state of the rock slope | Joint persistence [%] | Type of joint filling | Principal steps of the numerical analysis |
|---|---|---|---|---|
| 1 | | 30 | | Initial equilibrium |
| 2 | | 50 | | |
| 3 | All frozen (-4 to -5 °C) | 70 | no ice | |
| 4 | | 90 | | Progressive destruction of rock bridges |
| 5 | All frozen (-4 °C) | 100 | ice-filled | |
| 6 | 1st layer at -3 °C | 100 | ice-filled | |





| | | | |
|---|---|---|---|
| 7 | -2 °C | | |
| 8 | -1 °C | | Stepwise warming from rock surface to core in 4 steps |
| 9 | -0.5 °C | | |
| 10 | 1st layer unfrozen | | |
| 11 | 2nd layer unfrozen | | |
| 12 | 3rd layer unfrozen | 100 | thawed layers: no ice / frozen layers: ice-filled | Stepwise thawing from rock surface to core in 6 steps |
| 13 | 4th layer unfrozen | | |
| 14 | 5th layer unfrozen | | |
| 15 | All unfrozen | | no ice | |

1) Initial state: The numerical analysis was started by developing an initial equilibrium with a low joint persistence of 30 % (Model stage 1 in Table 6). The summit crest is assumed to be fully frozen representing an undefined moment in the past.

2) The progressive loss of rock bridges: The destabilisation of a permafrost-affected rock slope is initiated by the progressive destruction of cohesive intact rock bridges which prepare new shear planes along which displacement can take place (Krautblatter et al., 2013). The current overall joint persistence was assessed to 90–100 % as we observed a high trace length of the discontinuities and a displacement of several decimetres along the main shear zone indicating that most of the rock bridges are lost. As such, the progressive increase of the joint persistence from 30 to 90 % was implemented for the whole model by a reduction of the joint strength in three steps (Stages 1–4 in Table 6). For this, we decreased the apparent joint cohesion and friction angle for each model stage based on the approach by Jennings (1970) (Table 7). This procedure simulates "step-path" failure on the slope scale in a simplified way which is characterised by the interconnection of pre-existing, adjacent discontinuities through intact rock bridges (Camones et al., 2013; Eberhardt et al., 2004; Huang et al., 2015). The reduction in the shear strength involves processes of stress concentration, crack initiation, crack propagation and coalescence, slip weakening and the formation of a continuous failure plane or zone (Zhang et al., 2015), which were not numerically modelled as this would be beyond the scope of this study. The presented model of the Zugspitze ridge considers crack coalescence to occur by shear and tensile mode or a tensile-shear combination, which has been observed on the micro-scale by Zhang and Wong (2013). The resulting new connections can form coplanar or oblique to the pre-existing joints (Huang et al., 2015). At the scale of the Zugspitze ridge, we assume the joint sets K1, K3 and K4 to link in both coplanar and oblique form, leading to the formation of straight or stepped failure planes.

Joints may be partly filled with ice during the loss of rock bridges. However, according to Krautblatter et al. (2013), deformation and shear strength are mainly controlled by rock-mechanical processes at this early stage of destabilisation. As long as a certain part of intact rock bridges and asperities along joint surfaces are present, stresses are supposed to concentrate at these locations while the softer ice fillings may be squeezed away (in particular, in greater depths). As such, the assigned joint properties were those of ice-free joints. Following



Krautblatter et al. (2013), the advanced stage of accelerated displacements in a permafrost-affected rock slope begins as soon as most of the rock bridges are broken. Then, ice mechanics (fracturing of rock-ice contacts and ice, as well as the creep of ice) will increasingly replace rock mechanics (friction of rock-rock contacts and fracture of rock bridges) in controlling displacements along discontinuities and their potential acceleration. To introduce this stage of destabilisation, the joint persistence was further increased to 100 % and the frozen rock joints were filled up with ice (Stage 5 in Table 6). By doing this, we ensure the joint mechanical behaviour is fully controlled by ice mechanical processes. For simplification and due to data availability, the joint shear resistance was solely given by the fracture of ice or rock-ice contacts, while the creep of ice was neglected.

3) Warming permafrost: The next step was to gradually warm the frozen crest from the slope surface to the core from -4 to -0.5 °C (Stages 6–9 in Table 6). The spatial pattern of warming was based on the six rock layers defined in Fig. 5. The numerically modelled temperature range and the steps of warming were predetermined by the specific temperatures the implemented cohesion and friction angle of ice-filled rock joints are valid for (Eq. (8) and (9)). This temperature range is consistent with temperatures currently monitored in boreholes in permafrost rock walls across the European Alps (Gallemann et al., 2017; Noetzli et al., 2019). The warming procedure works as follows: Firstly, the outermost (orange) section was warmed to -3 °C, while inner sections remained at -4 °C (Stage 6). After that, the outermost section was warmed to -2 °C and the adjacent inner section (yellow layer in Fig. 5) was warmed to -3 °C, while inner sections remained unchanged (Stage 7). This procedure was continued until the outermost layer was at -0.5 °C (Stage 9). Each warming step was characterised by adjusting the temperature-dependent material parameters to the warmer temperature and by subsequent numeric cycling.

4) Thawing from the slope surface to the core was implemented in the same way as warming (Stages 10–15 in Table 6), until all subsurface layers of the summit ridge were unfrozen. The spatial pattern of thawing is illustrated in Fig. 5.

**Table 7: Implemented strength properties for ice-filled and frozen ice-free joints during the initial stage of rock bridge destruction, represented by a joint persistence of 30 to 90 %. The different cohesion and friction values are calculated after Jennings (1970) and depend on the estimated relative percentage of rock bridges and joints. The corresponding values at a joint persistence of 100 % are shown in Table 5.**

| Joint mechanical parameter | Type of model | Type of joint filling | Joint persistence [%] | | | |
|---|---|---|---|---|---|---|
| | | | 30 | 50 | 70 | 90 |
| Apparent cohesion [MPa] | Zugspitze summit crest (Sect. 4.1.) | Ice-free | 2.8 | 2.00 | 1.2 | 0.4 |
| | Simplified rock slope (Sect. 4.2.) | Ice-filled | -- | -- | -- | 1.0 |
| Apparent residual friction angle [°] | Zugspitze summit crest (Sect. 4.1.) | Ice-free | 42.1 | 40.8 | 39.5 | 38.2 |
| Apparent peak friction angle [°] | Simplified rock slope (Sect. 4.2.) | Ice-filled | -- | -- | -- | 31.9 |





To assess the level of a potential stability loss due to warming or thawing, we calculated the factor of safety (FS) by using the strength reduction technique. Thereby, the cohesion and frictional strength of the rock mass and the joints were reduced

simultaneously by varying strength-reduction factors in a series of simulations until failure occurred. A bracketing solution approach was applied to progressively reduce the bracket between stable and unstable solutions until it falls below a specified threshold (Itasca Consulting Group, 2019). The resulting FS is a single indicator of minimum stability which globally refers to the entire slope.

### 3 Sensitivity analysis for a simplified, warming permafrost rock slope

The relation between the dip of the joint sets, especially the bedding, and the slope-face is a crucial factor for rock slope stability (Cruden, 2003; Wyllie, 2018). Thus, we performed a sensitivity analysis on the numerical impact of varying (i) slope angles and (ii) orientations of the fracture network on rock slope stability transferring the Zugspitze model to a rock slope with a simplified topography and warming procedure without spatial differentiation. For this, the topography of the Zugspitze south-face was modified to a straight line. In accordance with the dimensions of the crest geometry, the height of the slope and the

width of the upper-face were standardised to 84 m and 32 m, respectively (Fig. 7). A stepwise warming of the frozen rock and ice-filled joints (Stages 5–9) was only applied to the study of the slope angle.

Modelling was applied to 12 different slope angles between 30 and 69°. For inclinations of 70° and higher, no initial equilibriums could be calculated as cycling exceeded the maximum computation time. Material properties and the joint set orientations were kept constant. In a first step, an initial equilibrium was calculated for a frozen rock slope with ice-filled joints

and a joint persistence of 90 %. In a second step, the joint persistence was increased to 100 %, and then the entire rock slope was warmed in four steps from -4 to -0.5 °C (equivalent to the procedure at the Zugspitze summit crest). Each warming step to the next degree centigrade was applied to the entire rock slope and not just to a single rock layer. Reducing the complexity of the model facilitates the transferability to other frozen rock slopes.

As mentioned above, we also remodelled the stability for varying dip angles of the fracture network with reference to the slope.

The positioning of the three model joint sets to each other was held constant, while rotating them counter-clockwise in steps of 15° for each of the 12 slope angles used for the first sensitivity test. 13 different orientations of the fracture network were modelled representing anaclinal and cataclinal slopes after Cruden (2003; Fig. 3). Hereafter, cataclinal slopes have a slope-face and bedding with the same dip direction whereas, for anaclinal slopes, the dips of both planes have the opposite direction. The orientation and dip of the fracture network was represented by the bedding and then applied to the scheme after Cruden

(2003). The simulated temperature was -4 °C.





## 4    Results

### 4.1 Numerical stability of the warming/thawing Zugspitze crest

At the initial state and during the stage of progressive rock bridge destruction (Stages 1–4 in Table 6, Fig. 6a–b), the entire
rock slope is still uniformly frozen at -4 °C. Here, the maximum block and zone displacements are very low and do not change

significantly (from 7.6 mm to 7.7 mm). The factor of safety decreases from 10.9 to 2.6. Filling the joints with ice and breaking
the remaining rock bridges does not cause important deformation or strength reduction (Stage 5 in Table 6). The maximum
block and zone displacements decrease by 0.02 mm.

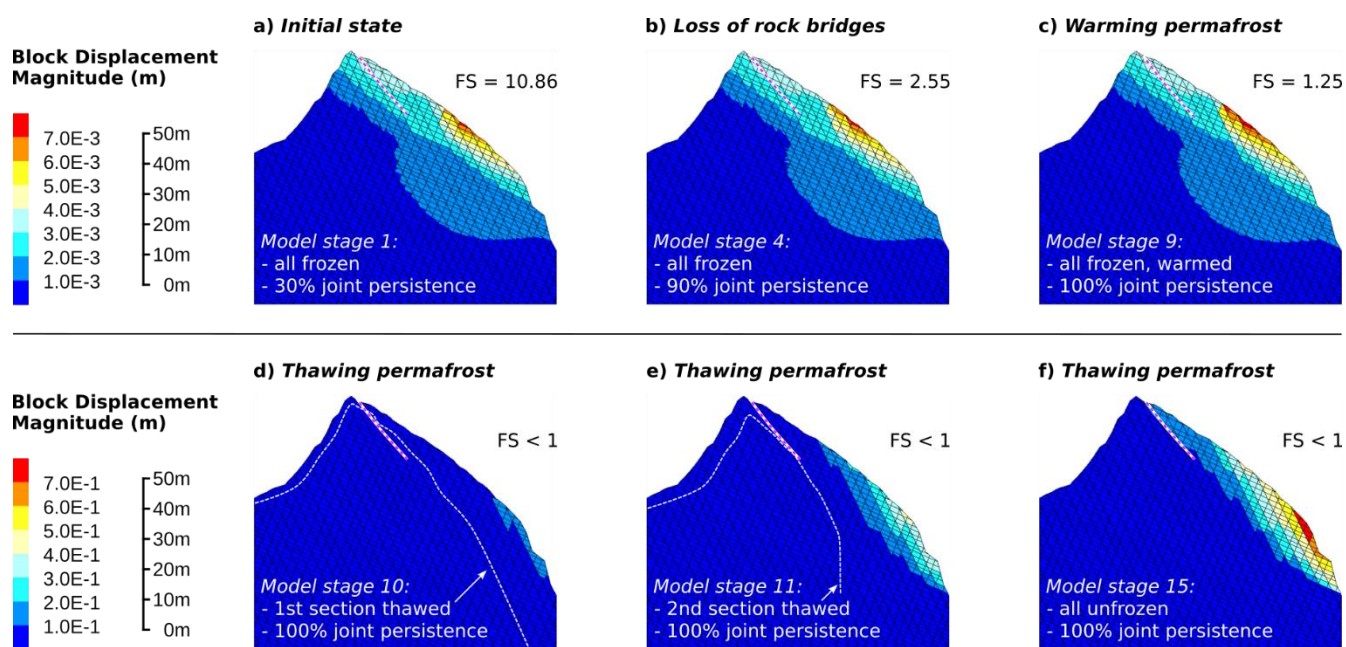

**Figure 6: Calculated spatial distribution and magnitude of displacements for the Zugspitze summit ridge. (a)–(c) Factors of safety
(FS) are given for Stages 1, 4 and 9 which refer to the loss of rock bridges and warming. (d)–(f) Stages 10, 11 and 15 correspond to
thawing with FS below 1. The prominent shear zone is marked by a white-red dashed line. The permafrost boundary in (d) and (e)
is highlighted by a white dashed line.**

The stage of stepwise warming was simulated by following the procedure outlined in Sect. 2.7, until the outermost rock layer
was at -0.5 °C and the two inner layers still remained at -4 °C (Stages 6–9 in Table 6). Stepwise warming was applied to the
rock layers defined in Fig. 5. Changes in the maximum displacements are very low and increase by 0.5 mm (from 7.7 to
8.2 mm, Fig. 6c). The spatial pattern of rock mass deformation does not change either. The factor of safety approaches unity
(FS = 1.25) but still remains above it.

Thawing firstly affects the surficial rock layer. As soon as thawing starts, the south-face becomes unstable and maximum
displacements increase to 141.1 mm (Stage 10, Fig. 6d). Rock deformations rise to 419.5 mm when the dimension of thawed





bedrock reaches the current state (Stage 11, Fig. 6e), and measure 776.3 mm for the whole rock slope thawed (Stage 15, Fig. 6f). The highest deformations (> 400 mm) concentrate on the upper 5–7 m of rock in the most inclined lower part of the slope (with a mean angle of 49°). The factor of safety lies below 1 for all Stages 10–15 which refer to thawing.


### 4.2 Sensitivity analysis for a simplified permafrost rock slope with rising temperature

Results of the studied numerical influence of a varying slope angle or orientation of the fracture network are presented in Sect. 4.2.1 and 4.2.2, respectively. The model domain is a rock slope with simplified topography. Warming steps are applied to the entire model domain without spatial differentiation.

**4.2.1 The influence of the slope angle on the numerical stability**

Figure 7 presents model results for exemplary slope inclinations between 60–64°. Here, the block displacements increase and the related FS decrease with higher slope angles and temperatures below the melting point. When a rock slope with a slope angle of 62° or more is warmed up, the factor of safety falls below unity at temperatures ≥ -1 °C. A 60° steep rock slope becomes unstable at temperatures ≥ -0.5 °C.






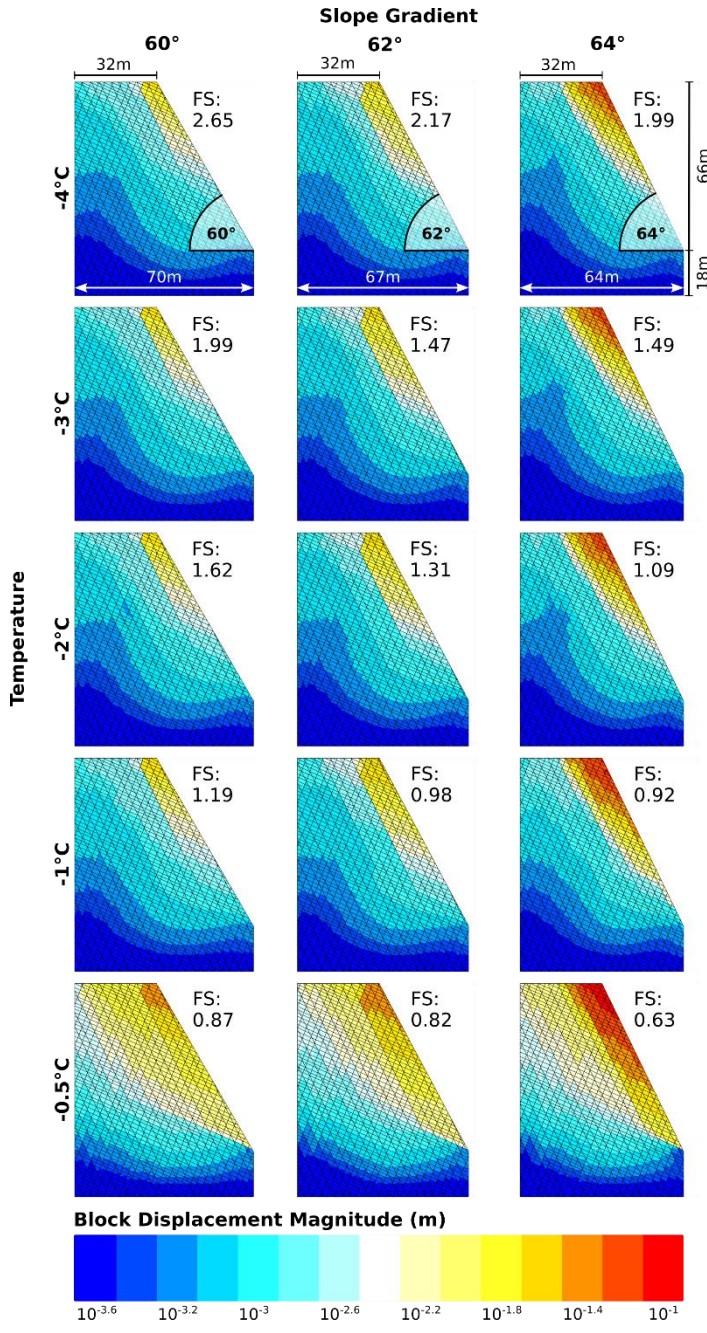

**Figure 7: Numerically calculated block displacements (on the log scale) for a simplified rock slope geometry with exemplary inclinations of 60°, 62° and 64°, and five temperatures between -4 and -0.5 °C.**

The relation between slope gradient, maximum displacements and temperature can be separated into three domains (Fig. 8a):



1)      Domain 1 (30 to 50°): The maximum displacements increase slightly by 0.4 mm per degree slope angle. The correlations for all temperatures are significant with p-values ≤ 4.4 %.

2)      Domain 2 (55 to 62°): Slope destabilisation sets in for -0.5 °C and above a slope angle of 50°. This becomes obvious due to the abrupt increase in displacements and the p-value > 5 %. For rock slopes at -1 °C and below, the onset of instability is visible above an inclination of 55°. All displacements are one magnitude higher than in Domain 1.

3)      Domain 3 (64 to 69°): The displacements increase suddenly for all rock temperatures with higher p-values. This points to a second stage of accelerating slope destabilisation. The displacements are one magnitude higher than in Domain 2.

The displacements are sensitive to the sub-zero temperatures at all slope gradients. However, the intensity of rock slope destabilisation is increased by higher rock temperatures close to the melting point (Fig. 8b): A warming from -4 to -2 °C leads to an increase in absolute displacements, being low in Domain 1 (0.2 mm °C$^{-1}$), but high in Domain 3 (7 mm °C$^{-1}$). A respective warming from -2 to -0.5 °C shows also an increasing trend with higher slope gradients, but is more pronounced with a massive increase from Domain 1 (0.6 mm °C$^{-1}$) to Domain 3 (45 mm °C$^{-1}$). Nevertheless, the highest temperature-dependent relative increase in displacements is observed in Domain 2 (Fig. 8b): Values increase from 9.2 % °C$^{-1}$ (Domain 1) to 38.5 % °C$^{-1}$ (Domain 2) and fall to 25 % °C$^{-1}$ for slopes steeper than 62° (Domain 3), which is valid for a warming from -2 to -0.5 °C. Again, the same pattern is less pronounced for a warming from -4 to -2 °C.





Earth **Surface**
**Dynamics**
Discussions

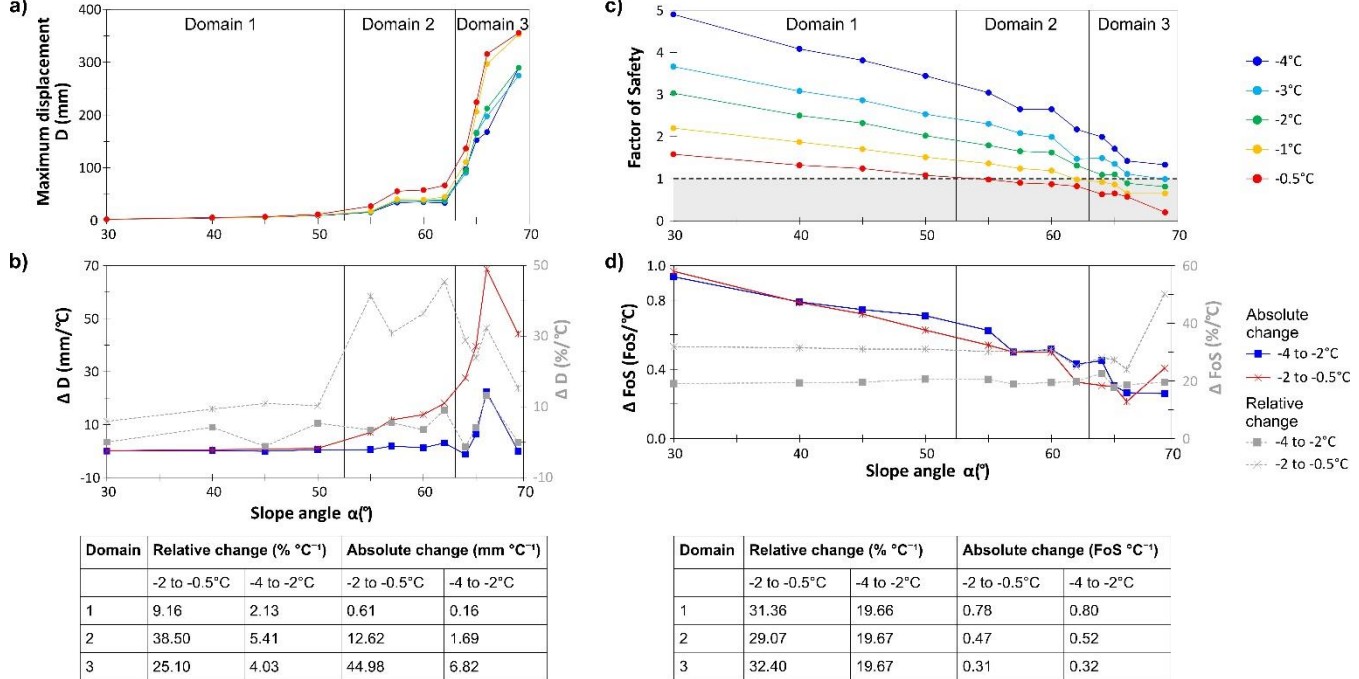

**Figure 8 provides critical slope angles and rock temperatures beyond which instability is introduced. (a) Calculated maximum displacements against slope angle for temperatures between -4 and -0.5 °C. The curves are assigned to three distinct domains dependent on the slope gradient: Above 50°, and a second time above 62°, the curves show a sudden onset of instability. (b) Absolute and relative increase in displacements versus slope angle for a warming from -4 to -2 °C and from -2 to -0.5 °C. (c) Calculated factor of safety (FS) against slope angle for different sub-zero temperatures. (d) Absolute and relative decrease in FS versus slope angle for a warming from -4 to -2 °C and from -2 to -0.5 °C.**

As could be demonstrated for the displacements, the factor of safety also depends on the slope angle and the temperature (Fig. 8c): For a rising steepness from 30 to 69°, the calculated FS decreases inversely and is always lower for higher temperatures. Warming from -4 to -0.5 °C reduces the FS of a permafrost rock slope by a mean factor of 3.3. The pattern of domains in Fig. 8a can be easily applied to the relation between FS, temperature and slope angle (Fig. 8c): The mean relative reduction of the FS per degree of the slope angle is low in Domain 1 (-1.8 % ± 0.4) and becomes higher over Domain 2 (-4.5 % ± 3.3), until it reaches a maximum in Domain 3 (-9.8 % ± 8.3). Correspondingly, rock slopes are stable for all modelled temperatures in Domain 1. When shifting to Domain 2, the FS falls below unity at -0.5 °C and at -1 °C which coincides with an abrupt increase in displacements. Rock slopes at -4 °C keep stable over the whole range of slope gradients.

As we did for the maximum block displacements, we tested if warming closer to the melting point led to a distinct diminishing pattern of the FS (Fig. 8d): The absolute decrease is very similar for both warming steps from -4 to -2 °C and from -2 to -0.5 °C. However, the relative decrease from -4 to -2 °C is always lower than the one at higher temperatures.



### 4.2.2 The influence of the fracture network orientation on the numerical stability

Modelling resulted in eight orientations of the fracture network that show a significant displacement acceleration and initiating slope instability above a critical slope inclination (Fig. 9a): For anaclinal slopes, the critical angles range between 50–62° and, for cataclinal slopes, they range between 55–62°. Cataclinal rock slopes with a bedding dip of 84 and 69° do not indicate a significant knickpoint in the displacement curves. As such, the corresponding critical slope angles were set to the maximum of the investigated values (69°). Fracture networks rotated counter-clockwise by 30° and 75° from the joint set of the Zugspitze

summit crest showed very noisy model data and were not used for the analysis.

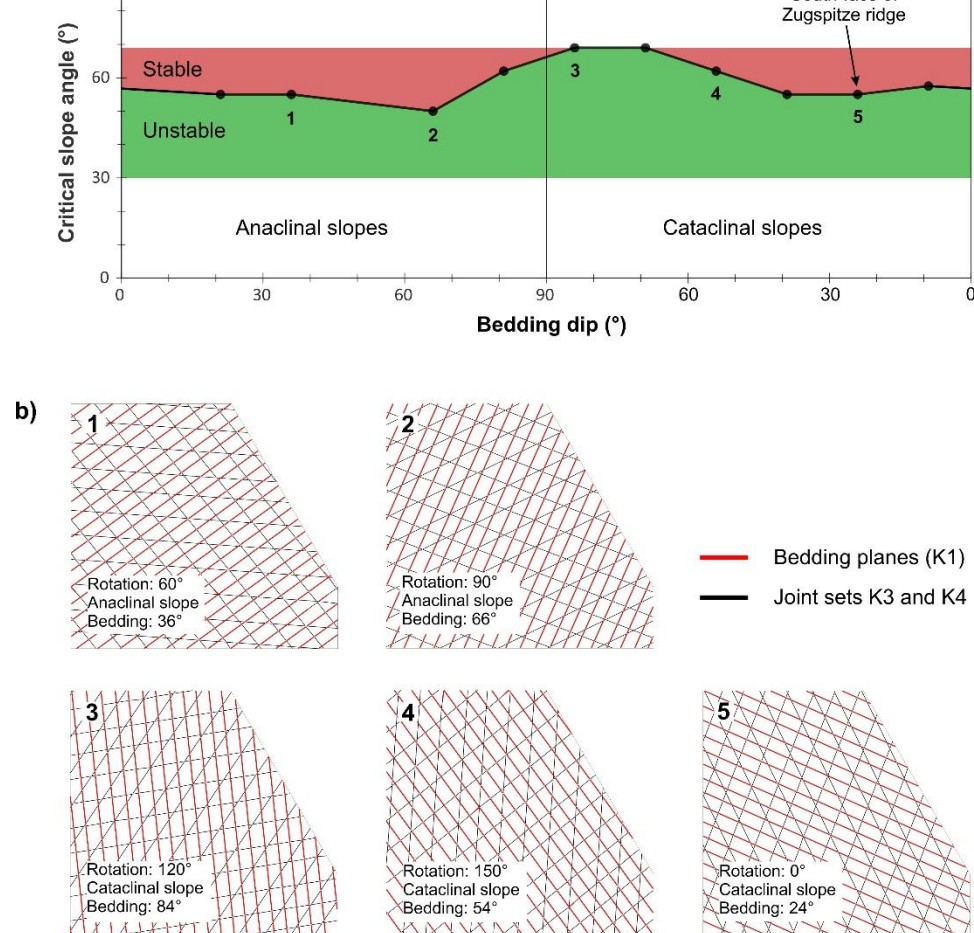

**Figure 9: (a) A rule-of-thumb for critical slope gradients and fracture network orientations of anaclinal and cataclinal permafrost rock slopes. For cataclinal slopes, both the slope-face and the bedding have the same dip direction, but for anaclinal slopes, the dips of both are oriented oppositely (Cruden, 2003). The green area refers to stable rock slopes while the red area refers to unstable ones.**
**For simplification, the fracture network is represented by the bedding dip. The coloured areas indicate the studied range of slope angles. (b) Five exemplary joint set constellations for cataclinal and anaclinal slopes with an inclination of 60°. The numbers of the rock slopes correspond to the numbers in (a). The joint spacing is ten-times bigger than in reality and is only applied for illustration. Example 5 refers to the Zugspitze summit crest.**






## 5    Discussion

The suggested modelling approach aims at supporting scientists and engineers in performing more accurate and realistic mechanical stability analyses by using temperature-corrected mechanical parameters with a temperature-dependent spatial assignment. It can be used for any permafrost-affected rock slope across the globe which is subjected to climatic warming.

The numerical model for the Zugspitze summit ridge successfully simulates the strength reduction at warming and thawing which was able to be demonstrated in the laboratory tests of this and of previous studies (Table 1). Warming of ice-filled joints from -4 to -0.5 °C takes the slope close to a critical level of stability (FS = 1.25). As soon as thawing sets in, the south-face becomes massively destabilised as the displacements increase by two orders of magnitude and the factor of safety falls below 1. While the warming of ice-filled joints below -0.5 °C does not initiate instability in the model, the ice loss at 0 °C can do it.

Destabilisation is triggered by a slight reduction of the joint cohesion from 0.18 MPa (at -0.5 °C) to 0 MPa (Table 5), even though the joint friction angle is increased from 13.5 to 29.3°.

Further, we show that destabilisation of a permafrost rock slope with ice-filled joints, a fracture network and rock/joint properties similar to the Zugspitze south-face can be initiated at a slope gradient > 50 or 55° (Domain 2 in Fig. 8: "First onset of instability") and significantly favoured above a gradient > 62° (Domain 3 in Fig. 8: "Accelerating slope destabilisation").

The transition from Domain 1 (Fig. 8: "Displacements increase slightly with the slope angle") to Domain 2 seems to be the most critical in terms of slope destabilisation, as it is characterised by the highest relative increase in displacements per degree of the slope angle. Interestingly, this could be confirmed by the study of the critical slope angle for varying orientations of the fracture network (Fig. 9): 50 % of the studied orientations have a critical slope angle of 50 or 55°, which corresponds to the transition from Domain 1 to Domain 2. 30 % of the orientations lead to instability with angles of between 57.5–62°, and 20 %

remain stable within the studied range of slope gradients.

The proposed procedure and the derived critical thresholds present useful tools to assess the stability of degrading permafrost rock slopes potentially endangering human life and mountain infrastructure. The critical stability thresholds regarding the slope angle, fracture network orientation and bedrock temperature can be used to detect high-mountain rock slopes which are susceptible to fail in the future. However, a prerequisite is to have data on the fracture network and the thermal field of the

rock slope under investigation. Even more a detailed stability assessment requires a number of site-specific input data, which is not realisable for a quick analysis of a current failure event.

The following sections provide a discussion on the numerical results of the site specific model (Sect. 5.1) and the model of the simplified permafrost rock slope (Sect. 5.2). After that, we discuss the limitations of the presented models (Sect 5.3) and if they can be transferred to other field sites in the Alps (Sect. 5.4).





## 5.1 The stability of the warming/thawing Zugspitze summit crest

The mean annual air temperature (MAAT) at the Zugspitze peak from 1991 to 2019 was -3.9 °C. It was 0.8–1.1 °C warmer than the three means of the prior 30-year reference periods between 1901 and 1990 (CDC-FTP-Server, Deutscher Wetterdienst). A model for the permafrost evolution at the Zugspitze peak by Gallemann et al. (2017) shows that between 1915 and 2015 the permafrost decreased by 2 m at the north-face and by 7 m at the south-face, with an increase in the permafrost temperature of approximately 1 °C. Until the end of this century, the mean air temperature at the Zugspitze peak is projected to increase by 3.2 °C, which can result in a total loss of permafrost at the Zugspitze summit (and the field site of this study) within the second half of the century (Gallemann et al., 2017). The corresponding scenario is represented by the last model stage 15 (Fig. 6f). Promoted by progressive thawing of the summit crest between the current state (Stage 11) and the second half of the century (Stage 15), the thermo-mechanical model predicts an ongoing increase in displacements which can prepare final slope failure.

The factors of safety for all stages with one or more thawed rock layers, including the current extent of permafrost (Stage 11, Fig. 6e), lie below unity which indicates slope instability. This is consistent with the observations of the current displacements and characteristics of the main shear zone at the field site. However, the high values calculated by the model are not comparable with the measured displacement rates at the field site (5.0 mm y$^{-1}$) for the following reasons: (i) The numerically calculated displacements are maximum values and refer to local sections in the model. (ii) Cycling in UDEC was not linked to specific time-steps. As such, we could not assign specific periods of time to the distinct, non-linear steps of warming or thawing. This did not permit us to estimate the approximate time period required for the calculated displacements to occur. (iii) Absolute displacements are path-dependent and affected by the history of calculations, including input and alteration of parameters as well as the stress-strain behaviour in the system.

Davies et al. (2001) postulate that the factor of safety of a permafrost affected rock slope is higher for unfrozen joints than for ice-filled joints between approximately -1.5 °C and zero degrees. In contrast to that, our numerical analysis of the thawing Zugspitze summit crest shows that many ice-filled fractures reach a minimum stability only upon thaw. This is indicated by the sudden and massive increase in displacements and the factor of safety falling below unity. The reduction in joint stability is significantly influenced by a related loss of the joint cohesion. The differing observations on the stability upon thaw may be explained as follows: (i) Our model assumes a lower joint roughness and uses the residual friction angle which is lower than the peak friction angle taken for the centrifuge model. We suppose that the simulated rock slope has already been moving for a long time and joint surfaces are flattened due to the progressive destruction of asperities along joints. In contrast, the sliding surface in the centrifuge model is not affected by any previous displacements. (ii) The calculated FS by Davies et al. (2001) corresponds to an independent moment in time, while the FS in the Zugspitze model is path-dependent and affected by the history of the previous numerical calculations.

Based on the 3D kinematic analysis (Sect. 2.3), we postulate that the most likely mechanism controlling the slope instability is a combination of a plane and a wedge failure. However, for the 2D numerical model and at the scale of the failure plane, we





consider various possible complex mechanisms: (i) Sliding of several rock blocks along a polygonal shear zone SZ1/SZ4 that is constituted by joint sets K1, K3 and K4. This complex type of failure is provided by antithetic fractures which enable shear
displacements between the moving blocks (Eberhardt et al., 2004; Poisel & Preh, 2004). (ii) Shear or tensile failure along joint sets K3 or K4 may favour shear displacements along K1 by supplying additional stress onto the blocks and inducing "step-path" failure (Camones et al., 2013; Huang et al., 2015). (iii) Failure at the scale of single blocks can be induced by forward block toppling with K1 and K3 or backward block toppling with K1 and K4 (Itasca Consulting Group, 2019). Both options will only work for higher columns subdivided by surfaces of K1 which stick together due to locally higher frictional or cohesive
strength.

The two joint sets K1 and K4, which daylight in the south-face, favour the displacements to concentrate on this slope aspect. However, the highest deformations are not coupled to the prominent shear zone. Instead they concentrate on the lower slope section with a mean gradient of 49°. This can be the result of (i) the dominating influence of the slope angle on the stability of the system and (ii) the lack of variation in the joint properties of the shear zone and the remaining joint sets (Table 5).

## 5.2 The stability of a simplified permafrost rock slope with rising temperature

The relevance of slope angles at the transition from Domain 1 to Domain 2 (Fig. 8), identified by the numerical model, is in accordance with documented rock slope failures in the Swiss Alps, adjacent areas in France and Italy (Fischer et al., 2012) and in the Mont Blanc Massif (Ravanel et al., 2010), which are mostly attributed to the degradation of bedrock permafrost: The mean slope gradient of the detachment zones in the Mont Blanc Massif was 54°, while the highest percentage of detachment
zones in the Swiss Alps and adjacent areas had slope angles of between 40–60°, leading to a postulated lower slope gradient threshold of 40–45° for rock slope failures.

We further demonstrate that the factor of safety falls below 1 when a rock slope with a gradient ≥ 50° is warmed from -4 °C to a temperature between -3 and -0.5 °C. This sensitive temperature range corresponds well to (i) the temperatures currently monitored in boreholes in permafrost rock walls in the European Alps (Gallemann et al., 2017; Noetzli et al., 2019) and (ii)
the temperatures which are characteristic for warm permafrost areas or the lower permafrost boundary (-5 to 0 °C). The latter posed the release zones for most of the rock slope failure events documented by Fischer et al. (2012) and Ravanel et al. (2010). The model results also demonstrate that the increase in displacements becomes more pronounced when approaching the melting point which indicates a non-linear relation (Fig. 8b): the relative and the absolute increase are always higher for a warming from -2 to -0.5 °C than for a warming from -4 to -2 °C. The relation between rising subzero temperature and reduced
stability was also observed by Davies et al. (2000, 2001). However, on the basis of our model results, we propose to complement the conclusions by Davies et al. (2001) in the following way: (i) The temperature of critical stability is highly dependent on the inclination of the main fractures versus the slope angle. (ii) It is of crucial importance to consider the stress conditions in a rock slope when extrapolating the results from the laboratory scale to the field scale. (iii) The use of rock instead of concrete samples for the laboratory determination of joint parameters results in a closer reproduction of the real conditions
along rock joints in the field.



### 5.3 Limitations and potential error sources

#### 5.3.1 Material parameters

Potential effects upon freezing, which can increase the rock mass strength, are mostly ignored in the GSI rock mass classification system. For instance, the suggested determination of the friction angle after Cai et al. (2004; Fig. 5) is based on
the temperature-independent Hoek-Brown constant $m_i$ and the *GSI* value, which could only be defined for the unfrozen rock mass currently exposed at the surface of the rock slope. As such, the unfrozen internal friction value had to be assigned to both the frozen and the unfrozen rock mass. Further, joint set specific parameters were not available, except for the joint normal and shear stiffness, which varied due to the different joint spacing (Eq. (6) and Eq. (7)).

We acknowledge that the presented model of the Zugspitze summit crest is simplified and limited by the knowledge of the
subsurface structure (i.e. joint spacing and joint persistence) and exact permafrost dimension. Especially the material parameters of the rock mass had to be estimated. For future models, we recommend developing studies in 3D to account for the missing joint set K2 and to provide a better impression of the kinematic behaviour of the rock slope. Further, we suggest to perform a detailed sensitivity analysis of the implemented material parameters in the future, which is beyond the scope of this paper.

#### 5.3.2 Numerical model

All model runs for the sensitivity analysis were started with a frozen rock slope at a temperature of -4 °C. However, if the warming of ice-filled joints is started at a temperature higher than -4 °C, model results may be different as material parameters with a lower strength have been assigned for the calculation of the initial equilibrium. To test this potential effect, we reiterated the same model and started warming at -2 °C. However, the same general pattern of dependency between maximum
displacements and slope inclination could be identified (Fig. S9). Furthermore, the displacement magnitudes for both model options did not differ either.

We acknowledge that the spatial warming pattern for the Zugspitze model is simplified and based on a static temperature field derived from ERT, rock thermistors and published thermal models (Böckli et al., 2011; Noetzli et al., 2007; Noetzli, 2008). For the future, we recommend developing a heat flow model of the crest for different time steps to better assess the warming
pattern for the coupled model. However, this is clearly beyond the scope of this article.

The number of calculated cycles for each warming step influences the available time for the rock slope to react to a change in stress-strain due to a modification of material parameters. The warming to the next degree centigrade was, by default, calculated with 3000 cycles. This was a compromise between a representative number of cycles for potentially reaching a numerical equilibrium (for stable rock slopes), and a reasonable, not too time-consuming calculation. Computations for
low/intermediate slope angles and temperatures reached an equilibrium prior to 3000 cycles, whereas calculations for higher slope angles and temperatures mostly failed to reach an equilibrium (for unstable rock slopes) or scarcely they required more than 3000 cycles. To prevent a potential cut-off effect of cycles that are relevant for a realistic stability analysis, all the





calculation steps were repeated with a higher number of 6000 cycles. The corresponding results show a very similar pattern as for the model runs with 3000 cycles (Fig. S10): (i) A significant acceleration of displacements above 50 and above 62° and (ii) higher displacements for temperatures closer to the melting point. The test confirms the reliability of the chosen number of cycles for each warming step.

The spacing was set five-times greater than in reality (Table 2) due to computational limits defined by UDEC. By doing this, reasonable computation times were achieved and the model still reproduced slope conditions close to reality. Nevertheless, we expect displacements to increase slightly as was shown by Gischig et al. (2011a), and which is an obvious effect due to a reduced rock mass strength (Cai et al., 2004; Fig. 7). Though, the comparability among all calculated model stages was not affected by the larger joint spacing as it was kept constant for all calculations.

The discontinuities were constructed fully persistent, although block displacements may have been favoured and velocities augmented along persistent joints. However, when implementing non-persistent joints, UDEC discards those which do not split a block. An irregular joint and block pattern may be formed which does not correspond to reality. To omit the potential result of favoured higher displacements and velocities, the joint strength parameters were increased in the first model stages following Jennings (1970).

To keep this first version of a thermo-mechanically coupled model as simple as possible, the numerical representation of the advanced stage of rock slope destabilisation was reduced to (i) the fracturing of ice and rock-ice contacts along frozen joints and (ii) the frictional strength of rock-rock-contacts along thawed joints. Future model codes should also include the preceding creep of ice at lower displacement rates.

## 5.4 Transferability and implications for other field sites

The general procedure for a combined thermo-mechanical stability model (Fig. 1) can be applied to any warming or thawing permafrost rock slope. However, the results of the sensitivity analysis with input data from the Zugspitze summit ridge (Sect. 4.2) are valid for rock slopes with increasing subzero temperatures, consisting of fractured limestone, with strength and deformability similar to the Wetterstein limestone tested in this study. A transfer to permafrost rock slopes with a different lithology is possible, but it requires more laboratory calibration tests and modelling. A provisional transfer may be possible with the following justification:

Mamot et al. (2018, 2020) studied the shear strength of ice-filled permafrost rock joints and developed a resilient temperature- and stress-dependent failure criterion. They postulate that it can be transferred to the majority of the rocks that are relevant for observed permafrost rock slope failures in the Alps. Since this failure criterion was used to calculate the cohesion and the friction of the ice-filled joints in the presented model, the numerical description of them may be applied to other lithologies, too. However, the joint stiffness and the mechanical parameters of the rock mass still vary among different rock types and may lead to different model results. To get a first impression of this potential effect, we performed a couple of model test runs with varying values for the mechanical properties of the rock mass ranging from very low to unrealistically high. As the displacements remained within the same order of magnitude (Fig. S11), we infer that the model results will not be significantly



influenced by other rock types with distinct properties of strength and deformability. Nevertheless, we strongly recommend to verify the effect of different rock types by thorough modelling in the future.

Further, it is important to consider that the results in Fig. 9 are valid for a specific constellation of three joint sets and not only for the bedding, as presented by Cruden (2003). Hence, the results can only be applied to rock slopes with a similar joint set

configuration as in this article.

Considering the constraints above, the simplified model, developed by the sensitivity analysis (Sect. 4.2), can be transferred to warming permafrost rock slopes with ice-filled discontinuities and slope gradients smaller than 70°. In addition, these rock slopes consist of limestone (and probably mostly all rock types relevant for permafrost rock slope failures in the Alps), and they contain three joint sets separated from each other by an angle of 45°.

**6    Conclusion**

The warming climate in high mountains is expected to lead to a higher number of rock slope failures influenced by progressive permafrost loss (Gobiet et al., 2014; Huggel et al., 2012). To tackle the challenge of assessing the stability of degrading permafrost rock slopes in a warming climate, we need to understand how rock-ice mechanical components control rock slope destabilisation and how failure close to melting, and at thawing, can be mechanically expressed in models. For this, we have

developed the first model which is capable of supporting scientists and engineers in performing mechanical stability analyses of warming or thawing permafrost rock slopes. In this context, we provide (i) a universal procedure for the input of thermal and mechanical data, the model setup and the modelling strategy, (ii) a numerical application to a specific test site at the Zugspitze peak, and (iii) the first numerically derived critical stability thresholds related to the rock mass temperature, slope angle and orientation of the fracture network. The related main findings are summarised as follows:

(i)–(ii) The proposed instruction for the set-up of a coupled thermo-mechanical stability model (Fig. 1) can be used for any permafrost-affected rock slope across the globe which is subjected to climatic warming. However, it requires a number of various input data which are to be collected in the field and in the laboratory. The numerical modelling approach was successfully applied to the permafrost-affected Zugspitze summit ridge: we used rock specimens picked from the Zugspitze summit area and performed a large number of mechanical laboratory tests on various

740         frozen and unfrozen intact rock and rock joint properties. Intact rock properties were converted to rock mass characteristics using internationally standardised mathematical equations. The topography of the crest, the dimensions of the unstable rock mass, the potential failure mechanism, large-scale shear zones and the geometrical properties of the fracture network, fracture displacements as well as mechanical rock mass and rock joint properties were determined due to geotechnical methods (mapping of the fracture network, Schmidt hammer, Barton comb,

745         digital tape extensometer) and UAV-based photogrammetry. ERT and logging of near-surface rock temperatures were applied to identify current frozen and unfrozen slope sections, to assign frozen or unfrozen rock mass and joint parameters to the corresponding model sections and to derive a rough spatial warming pattern. The modelling





procedure was divided into three principal stages: rock bridge destruction, warming and thawing. Since process-specific and temperature-dependent input parameters were changed when switching from one stage to the next, the Mohr-Coulomb plasticity constitutive model is capable of accounting for all processes which act during destabilisation.

The model demonstrates that the factor of safety decreases to a critical level of stability when rock bridges break and frozen rock and ice-filled joints gradually warm from -4 to -0.5 °C. Rock slope failure is introduced upon thawing of the outermost rock layer as the factor of safety falls below 1. Permafrost at the Zugspitze summit is projected to be completely lost within the second half of the century as a result of climatic warming (Gallemann et al., 2017). The Zugspitze model predicts an increase in displacements upon full thaw of the summit crest potentially leading to final slope failure.

(iii)  The critical thresholds were calculated in the context of a sensitivity analysis of the Zugspitze model with a simplified geometry and warming pattern. The simplified model demonstrates that the displacements increase with the slope angle and the factor of safety decreases inversely. Rock slope failure is initiated at slope angles of greater than 50° and is even more intense above a slope gradient of 62°. The critical slope angle varies between 50 and 62° depending on the orientation of the fracture network versus the slope face. These calculated results correspond well to inventories of rock slope failures in permafrost areas in the European Alps.

Further, the generalised model shows that warming from -4 °C to a temperature between -3 and -0.5 °C initiates rock slope instability for rock slopes ≥ 50°. The increase in displacements intensifies closer to the melting point. The identified sensitive temperature range between -3 and -0.5 °C is consistent with the one currently measured in boreholes in permafrost rock walls across the European Alps; further, it is characteristic for warm permafrost areas which involve the release zones for most of the failure events documented in the Swiss, Italian and French Alps.

The derived critical thresholds can be applied to warming permafrost rock slopes with (a) ice-filled joints, (b) limestone equivalent to Wetterstein limestone, or probably most of the rock types relevant for permafrost rock slope failure in the Alps, (c) slope angles smaller than 70° and (d) various orientations of the fracture network consisting of three joint sets. Further modelling and laboratory tests are required to complete this first approximation of an application to other field sites.

Considering the constraints above, the critical stability thresholds in terms of the slope angle, fracture network orientation and bedrock temperature can be used to detect high-mountain rock slopes which are susceptible to fail in the future. Since we are still at the beginning of developing numerical models for the mechanical response of a rock mass to warming/thawing, there is a huge potential for improvements. For instance, time-dependent cycling is not yet implemented in most of the numerical models, neither it is in the presented models. Hence, they cannot be used to forecast a precise time of rock slope failure. The time of the final accelerated slope failure is coupled to the onset of thawing bedrock permafrost, which varies among field sites and has to be estimated by a separate thermal model.



*Code/Data availability.* All obtained data from the model, laboratory tests, geotechnical and geophysical surveys in the field as well as the numerical code will be freely accessible in the public data repository https://zenodo.org/. Data archiving is underway and will be finished within a short time. In the meantime, the data is available via the enclosed zipped folder enclosed
in the supporting information.

*Author contribution.* Philipp Mamot (PM), Samuel Weber (SW) and Michael Krautblatter (MK) developed the general modelling approach. PM performed the geotechnical and geophysical surveys at the study site and analysed the related data. The setup and the procedure of the laboratory experiments was designed by PM and MK. The rock samples were prepared by
Saskia Eppinger (SE) who also executed the laboratory tests and analysed the data, supported by PM. PM developed the numerical discontinuum code with UDEC and performed modelling. The modelling strategy was designed by PM, MK and SW. Analysis of the model results was done by PM and SW. The manuscript was prepared by PM, with a substantial contribution from SW and MK.

*Competing interests.* The authors declare that they have no conflict of interest.

*Acknowledgements.* The work for this article was funded by the Technical University of Munich. The authors thank the colleagues of the Chairs of Landslide Research and Engineering Geology of the Technical University of Munich as well as all motivated students who either supported the realisation of the laboratory campaigns or helped to perform the field surveys
at the Zugspitze. However, the authors are particularly grateful for the valuable contribution of their colleagues Benjamin Jacobs, Andreas Dietrich, Riccardo Scandroglio, Bettina and Florian Menschik, who helped to collect and to analyse the data. Finally, a thanks goes to the Fritz and Lotte Schmidtler Foundation for financing the TUFF fellowship that was held by Samuel Weber.

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
