# Peer review of "A temperature-dependent mechanical model to assess the stability of degrading permafrost rock slopes"

_Earth Surface Dynamics, 2020_

## Referee Comment (RC1) · Anonymous Referee #1 · 3 Dec 2020

The manuscript by Marmot et al. presents a failure model for degrading permafrost rock slopes, which couple thermal and mechanical properties of the slope. They use the Zugspitze crest (Germany) as an example, from which they draw general conclusion about the thermo-dynamical behaviour of steep rock slopes in permafrost environments.

In general, the manuscript is mostly well written and addresses a topic of wide scientific interest, and therefore deserves publishing in an international journal. They demonstrate the stability in steep rock faces is influenced of ground temperatures, and stability is reduced with increasing temperatures toward the melting point. I am not

expert in some of the details of the mechanical modelling used, so I do not have any comments of the parametrisation and mechanical equations. This should be addressed by a reviewer with proper competences within this field. It is obvious, that the authors present innovative results by combining the ground thermal regime and mechanical behaviour in a real slope setting, at least I am not aware of many similar studies. It has to be mentioned here that this topic is of outmost interest in the light of passed and predicted atmospheric warming. However, before possible publishing I would suggest some issues to be solved:

- I struggle a bit with the structure of the manuscript. The authors want to show the result of a rock mechanical model, where mechanical parameters change with changing ground temperatures. This is in principle fine. However, chapter 2 is not only the mechanical model, but also a lot of results which serve as important input to the model, and to a large degree documented as supplemental. I would suggest to re-structure this part, distinguishing the site description, rock joints and kinematics, lab work and permafrost distribution from the rock mechanical set up and modelling. Here I assume that the first topics I mentioned are derived in this manuscript, as many of these points are more explained in the supplemental. The authors want to present a large material, from the present state of permafrost and stability of the Zugspitze crest, to the development of a thermo-mechanical model, resulting in for me interesting conclusions.

- The ground temperature distribution and temporal change is a crucial part of the manuscript. The authors start with a thermal setting derived from ERT measurements, which were calibrated in the laboratory in terms of temperature and associated resistivity. Much of the underlying assumption are presented in the supplemental, on has to jump back and forth to find the relevant information. Maybe it is wise to include some of it in the main document. It is positive that the authors discuss the possible inclusion of a heat-flow model to assess the geometry of the permafrost body in the Zugspitze crest.

- The manuscript is lengthy and partly full of redundancies/summaries and can be

shortened substantially and with this make space for inclusion of some of the supplementary information. E.g. the conclusions are too long, and partly either a summary or even a discussion (see below). Make a list of conclusions of your work and avoid lengthy summaries.

The following minor comments apply:

l. 22: last sentence abstract, is this so? Only to check. . ..

l.79: Randa rock slope (Gischig et al) – no permafrost there?

l. 103: Chapter 2, see comment above. This is a mix of methods and results. I would suggest you first evaluate the setting (rock joints, kinematics, thermal regime) and then make a model based on the settings.

l. 116: Do not understand the sentence, velocity are reduced 84% during summer? Please clarify. Fig. 2: Show first the map (setting), then the images. Scale bar is missing. Maybe use colors for elevation, or use a normal topographic-type map. DEM is of no interest here.

l. 161: You give "results" in the setting chapter. . ...

l. 177: all 2.4. is a result of investigations done for this manuscript (?)

l. 225: 2.5. is a new part, and should probably be handled as such.

l. 339: Why is this an own chapter? This is part of the modelling?

l. 454: Avoid all sentences starting "as mentioned above. . .".

Fig. 7: Consider to give these displacements in mm or so.

l. 557, discussion: This all start is a totally uncecessary summary. You can delete the whole page and start with 5.1.

652: The discussion should probably start with the limitations etc. It destroys a bit the flow of the discussion,

l. 725: The conclusions are far too long. 50% of the conclusions is a summary, in addition to some discussion points (e.g. l. 767). You can avoid repeating stuff and citing literature in the conclusion. You should show what your study has come up with.

---

## Referee Comment (RC2) · Anonymous Referee #2 · 22 Dec 2020

"Stability assessment of degrading permafrost rock slopes based on a coupled thermos-mechanical model" by Mamot et al.

The manuscript reports on the influence of temperature changes on the stability of permafrost affected rock walls. The authors combine rock mechanical and geophysical investigations and numerical studies to explain temperature induced destabilization of deep-seated rock slope instabilities. The study is based on a case study in South Germany (Zugspitze). The paper provides interesting insight to the permafrost situation at the studied site and a series of new laboratory results. However, the paper contains a series of major flaws suggesting that the paper cannot be published in its current form.

[Figure]

The modelling approach used in this study is an uncoupled approach and thus the title and text are misleading. Regions with anticipated temperature and rock property changes are simply assumed and not modelled. While adjusting rock properties based on temperature changes is an appropriate assumption, a thermo-mechanical coupled approach (as claimed in the title and many times in the text) also requires modeling temperature propagation. Transient temperature and rock property changes may lead to completely different results and may also modify the hydraulic situation. A strongly simplified approach, as used in this paper, needs to be confirmed by truly thermo-mechanical coupled numerical simulations. The required confirmation has not been provided. Thus, the validity of the approach is highly questionable. The black line in Figure 2b indicates the modelling cross section. The cross section ends in the steep rock slope south of the ridge. Thus, the modelling domain is certainly too small and boundary effects must be anticipated. A validation of the size of the model domain is not provided, but mandatory. In addition, the kinematic analysis suggests a combined planar sliding and wedge failure mode. What is dominant? Wedge failure cannot be analyzed in 2D models. Consequently, the results are unreliable. The author's state that many of the above points are under considerations (chapter 5.3) but clearly beyond the scope of this paper. Unfortunately, I disagree. It is mandatory to investigate these issues before claiming a general approach.

Additional list of major issues: The term "fracture displacement" is used in both the main body of the paper and the supplementary material. From the text it seems that linear strains (fracture opening) have been measured and no displacements (neither relative nor absolute displacements). The factor "D" in the GSI approach was set to zero, because no blasting was performed. Although the GSI approach is highly uncertain, it is clearly stated that in unstable rock slopes "D" equals 1. The derivation of rock mechanical properties for both intact rock and rock mass remain unclear and uncertain. Many rock properties are derived indirectly, although laboratory tests have been performed, which allow a direct determination of static rock properties (e.g. Young's modulus and Poisson's ratio, the normal/shear stiffness and shear resistance of fractures). The primary aim of the model is on stability considerations, where elastic properties are typically irrelevant. This can easily be shown in a simple numerical study. Instead, assumptions are made for defining elastic properties, which are not valid. For example, it is assumed that the ratio E/G is analogous to kn/ks. There is no theoretical justification for such an assumption (this is also stated in the cited paper). Saturation plays a major role on how the rock properties changes with changing temperature. I strongly doubt that assuming saturated rock mass conditions in a steep alpine ridge is valid. The natural saturation degree should be measured before performing laboratory tests on frozen samples and should be documented for all test specimens. The influence of the saturation degree and loading rate on the rock properties needs to be analyzed.

---

## Author Comment (AC1) · 27 Dec 2020

We thank the two anonymous referees for reviewing our submitted manuscript and writing helpful and interesting comments. However, we would like to shortly respond to some comments of the evaluation of the second referee. While doing so, we will focus on those modelling assumptions which we think are essential:

(i) Saturated frozen condition: We assumed the summit crest to be saturated for the following reasons: Sass (2005, ESPL) and Rode et al. (2016) show that the rock moisture in depths greater than 15 cm does not fluctuate (which occurs near the rock surface due to meteorological influences like precipitation, wind or insulation). Deeper than 15 Printer-friendly version

cm, the rock saturation ranges between approximately 75 and 90 % dependent on the rock type and its porosity. In addition, water saturated rock samples have to be used to calibrate the electrical resistivity of rocks for ERT in rock walls (Krautblatter et al., 2010). The DC conductivity of porous or fractured subsurface matter highly depends on saturation of the pore space and conductivity of the pore fluid (Supper et al., 2014). Only saturated rock samples show a significant difference in conductivity/electrical resistivity when switching from the unfrozen to the frozen state, or vice versa (see also Mellor, 1973). The ER inversions of the presented study or many others in permafrost bedrock confirm that the rock mass under investigation is saturated, since big differences in the ER-values are detected.

(ii) Coupled modelling approach: We interpret the term "coupled thermo-mechanical model" differently. We refer to the fact that we for the first time include frozen and unfrozen bedrock material properties which are assigned to specific regions in the model and changed due to warming. However, we suggest to rename the title into "A temperature-dependent mechanical stability model for degrading permafrost rock slopes", and to adjust the wording in the text.

(iii) General approach: The presented general approach for the set-up of the model and the modelling procedure (Fig. 1) is formulated very generally. It can be easily applied to any degrading permafrost rock wall (if considering the model is temperature-dependent and not a coupled one, and hence, does not require numerical thermal modelling). We proposed one specific way how the set-up of the model and the modelling procedure can be applied to the Zugspitze summit crest. For instance, the conversion of the intact rock properties to those of the rock mass can also be performed by other techniques, like the Q-value.

(iv) Failure mechanism: Planar sliding is clearly the dominant failure mechanism at the site, though it is a combination with a wedge failure. In addition, the transects for the geophysical measurements and the profile for the numerical model run through the stepped planar failure part of the unstable rock mass (shear zones SZ1, SZ4, SZ5; left

**ESurfD**
part in Fig. S2). We can make this clear by adding a comment in the text.

(v) Modelling domain: The numerical analysis was applied to a locally constrained section of the rock slope for the following reasons: (i) The selected model domain fully incorporates the unstable rock mass under investigation. (ii) It perfectly accounts for the spatial conditions of the rock-ice-mechanical processes we want to simulate. (iii) The upper and middle sections of the Zugspitze south slope do not contain significantly lower gradients as in the model domain. In contrast, the talus slope is flatter. However, it starts approximately 200 m below (at 2650 m a.s.l.), and ends in the Zugspitzplatt at ca. 2500 m a.s.l. We expect no big differences in the model results for a bigger size of the model domain, since the slope gradient downslope does not change significantly. However, we will test the potential effect of a bigger model domain size due to additional model test runs.

(vi) Elastic properties: It is true they are not the most important values for the presented numerical calculation, but as we had laboratory values for them we decided to better include them.

(vii) Disturbance factor: The disturbance factor D is set to values higher than 0 if the slope under investigation is affected by significant blast damage. To our knowledge, a standardised recommendation of how to set D for an unstable rock slope without blast damage does not exist so far. Such as recommendation should be numerically validated and include a classification of various degrees of instability. As a consequence, we decided to stick to D = 0, as any other value seems to be arbitrary.

(viii) Fracture displacements: No major openings have been observed along the fractures, so the major measured elongation of crackmeters went into displacements.

The presented model is the first numerical attempt to assess the mechanical response of a permafrost-affected rock slope to warming/thawing. As a consequence, it is impossible to account for all relevant input parameters and possible influences in the first study/article. It is certainly necessary to include hydrostatic pressures or a heat flow **ESurfD**
model, and to perform more laboratory tests on mechanical properties in the future.

**ESurfD**

---

## Author Response (AR1)

**Final author response**

Dear editor,

we thank you very much for extending three times the deadline for the submission of the final response. Unfortunately, the preparation time for the final response has massively increased due to an exceptional situation caused by Corona. Illnesses have come along. The provided additional time gave us the opportunity to fundamentally revise this manuscript and to respond to the comments of the two referees.

Motivated by their helpful and constructive comments, we revised the manuscript in terms of structure (Sect. 2, Sect. 5 and supplements) and contents. Wherever possible, redundant information was removed and the manuscript was reduced to relevant information. Additional modelling was performed to analyse the effect of (i) a bigger size of the model domain and (ii) different values of the disturbance factor D on the mechanical response of the rock slope (Sect. 5.1, Sect. S7.2). Further, we shifted some parts of the supplements to the main text to reduce switching between these two texts. Some figures were unified (Fig. 1 and Fig. 4) and their graphical quality was improved (Fig. 8).

Enclosed you find the author response to the two revisions and the main article and the supplements with and without track changes.

Philipp Mamot, on behalf of all co-authors

**Response by the authors to the evaluations of referee 1:**

Dear anonymous referee,

we thank you very much for your constructive and valuable suggestions and comments related to the content and the structure of the submitted manuscript. We carefully addressed all of them and listed the changes in detail hereafter.

Philipp Mamot, on behalf of all co-authors

RC = Referee comment

AR = Author response
* * *
**RC:** "I struggle a bit with the structure of the manuscript. The authors want to show the result of a rock mechanical model, where mechanical parameters change with changing ground temperatures. This is in principle fine. However, chapter 2 is not only the mechanical model, but also a lot of results which serve as important input to the model, and to a large degree documented as supplemental. I would suggest to re-structure this part, distinguishing the site description, rock joints and kinematics, lab work and permafrost distribution from the rock mechanical set up and modelling. Here I assume that the first topics I mentioned are derived in this manuscript, as many of these points are more explained in the supplemental. The authors want to present a large material, from the present state of permafrost and stability of the Zugspitze crest, to the development of a thermo-mechanical model, resulting in for me interesting conclusions."

**AR:** As suggested by the referee we restructured Section 2 to clearly separate the part of the model input parameters from the model setup and the modelling strategy:

2. A general approach applied to the Zugspitze summit crest

      2.1 Characterisation of the mechanical and thermal setting

            2.1.1 Model profile

            2.1.2 Rock joint geometry and kinematic analysis

            2.1.3 Spatial permafrost distribution

            2.1.4 Mechanical properties

      2.2. Model setup

      2.3 Procedure for the numerical analysis

Furthermore, we shifted some of the relevant information on the study site, the kinematic analysis and the geophysics from the supplements to the main manuscript. In this way the reader does not have to switch between the two documents so often. For example, (l. 90):

*"2. A general approach applied to the Zugspitze summit crest*

[revised manuscript text omitted]

…"

**RC:** "The ground temperature distribution and temporal change is a crucial part of the manuscript. The authors start with a thermal setting derived from ERT measurements, which were calibrated in the laboratory in terms of temperature and associated resistivity. Much of

the underlying assumption are presented in the supplemental, on has to jump back and forth to find the relevant information. Maybe it is wise to include some of it in the main document."

**AR:** See author response above.
* * *
**RC:** "The manuscript is lengthy and partly full of redundancies/summaries and can be shortened substantially and with this make space for inclusion of some of the supplementary information. E.g. the conclusions are too long, and partly either a summary or even a discussion (see below). Make a list of conclusions of your work and avoid lengthy summaries."

**AR:** The **e**ntire manuscript was shortened wherever possible. For instance, the conclusion was changed to a list of concluding points which are more easy to read. Hereby, we avoided to summarise or to discuss (l. 670):

*"6. Conclusion*

*This study presents the first numerical model which is capable of performing comprehensive mechanical stability analyses of degrading permafrost rock slopes. In this context, we provide (i) a universal procedure for the input of thermal and mechanical data, the model setup and the modelling strategy, (ii) a numerical benchmark application to a specific test site at the Zugspitze peak, and (iii) the first numerically derived critical stability thresholds related to the slope angle, rock mass temperature and orientation of the fracture network. The related main findings are summarised as follows:*

*(i)     The temperature-dependent mechanical stability model can be used for any permafrost-affected rock slope across the globe which is subjected to climatic warming.*

*(ii)    Laboratory tests and field reconnaissance of the benchmark site Zugspitze exemplify thermal, geometrical and mechanical input data for the numerical model.*

*(iii)   Frozen and unfrozen bedrock material properties were assigned to specific sections in the model and changed due to warming/thawing. The modelling procedure was divided into three stages: rock bridge destruction, warming with ice-filled joints, and thawing. Process-specific and temperature-dependent input parameters were modified when switching from one stage to the next.*

*(iv)    The Zugspitze model demonstrates a stability decrease towards a critical level as a result of (a) rock bridge destruction and (b) gradual warming of frozen rock and ice-filled joints from -4 to -0.5 °C. Surficial rock slope failure starts coincident to thawing of the outermost rock layer. Upon full thaw of the summit crest, expected within the next five decades, the model predicts an increase in displacements which potentially lead to final slope failure.*

*(v)     We developed a framework to generalise and upscale the Zugspitze model: Critical stability thresholds were calculated in the context of a sensitivity analysis of the Zugspitze model with simplified geometry and warming pattern.*

*(vi)    The dependence between instability and the slope angle can be classified into a stable first domain (≤ 50°), a second domain with a first onset of instability (55– 62°), and a third domain characterised by an accelerated slope destabilisation*

*(≥ 64°). The greatest relative increase in displacements is observed in the second domain.*

(vii) *Warming from -4 °C to a temperature between -3 and -0.5 °C initiates instability for rock slopes ≥ 50°. Destabilisation is more pronounced for warming closer to the melting point (from -2 to -0.5 °C) than for warming from -4 to -2 °C. This difference becomes greatest in the second domain.*

(viii) *For anaclinal slopes, the critical slope angles range between 50–62°, and for cataclinal slopes, they range between 55–62°.*

(ix) *The calculated critical slope angles and rock mass temperatures correspond well to the characteristics of documented rock slope failures in permafrost areas in the European Alps, which often showed large amounts of residual ice in their scars.*

(x) *The critical thresholds can be applied to warming permafrost rock slopes with (a) ice-filled joints, (b) limestone equivalent to Wetterstein limestone, or probably most of the rock types relevant for permafrost rock slope failure in the Alps, (c) slope angles smaller than 70° and (d) various orientations of the fracture network consisting of three joint sets.*

(xi) *The critical thresholds can be used to detect rock slopes which are susceptible to fail in the future and potentially endanger human life and mountain infrastructure. For this, it is a prerequisite to have data on the fracture network, lithology, geometry and the thermal field of the investigated rock slopes. In contrast, a detailed stability assessment of a single rock slope requires a number of further site-specific input data.*

*We are at the beginning of developing numerical models for the mechanical response of a rock mass to warming/thawing. Therefore, there is a huge potential for improvements. For instance, more mechanical laboratory tests are necessary to improve process understanding and to extend the number of frozen and unfrozen material parameters for modelling. Future numerical models should be based on further sensitivity analyses, or include water pressure or the creep of ice at lower displacement rates within the advanced stage of rock slope destabilisation. Time-dependent cycling is not yet implemented in most of the numerical models, neither it is in the presented ones. Hence, they are not suited to forecast a precise time of rock slope failure. Further, the spatial warming pattern for the Zugspitze model is simplified and based on a static temperature field derived from ERT, rock thermistors and published thermal models. For a more accurate characterisation of the warming pattern, future mechanical models should include a subsurface temperature field based on a combination of geophysics and a heat flow model for different time steps."*
* * *
**RC:** "l. 22: last sentence abstract, is this so? Only to check…"

**AR:** To our knowledge, it is the first mechanical model. However, we modified the sentence as we are not totally sure about that (l. 22):

*"…Here we present a benchmark model capable of assessing the future destabilisation of degrading permafrost rock slopes."*
* * *
**RC:** "l.79: Randa rock slope (Gischig et al) – no permafrost there?"

**AR:** The authors of Gischig et al., 2011a, 2011b and 2011c (see References in the main manuscript) do not report on the observation of permafrost at this site. The low altitude may exclude the occurrence of permafrost.
* * *
**RC:** "l. 103: Chapter 2, see comment above. This is a mix of methods and results. I would suggest you first evaluate the setting (rock joints, kinematics, thermal regime) and then make a model based on the settings."

**AR:** See author comment above. The evaluation of the thermal and mechanical setting at the study site is given in Section 2.1, while the setup of the model and the modelling procedure are presented in Sections 2.2. and 2.3, respectively.
* * *
**RC:** "l. 116: Do not understand the sentence, velocity are reduced 84% during summer? Please clarify."

**AR:** To make the sentence clearer we modified it as follows (l. 175):

*"Measured fracture displacements between 2013 and 2019 show that the rock mass creeps slowly at an average of 2.1 mm yr$^{-1}$ (Fig. S3). Between October and May, deformation rates are 84 % lower than in the remaining summer months."*
* * *
**RC:** "Fig. 2: Show first the map (setting), then the images. Scale bar is missing. Maybe use colors for elevation, or use a normal topographic-type map. DEM is of no interest here."

**AR:** As suggested by the referee, we exchanged the overview map and included a scale bar (see figure below). In the new Fig. 1 (l. 108) of the revised manuscript, the overview map is presented at the top and it is zoomed in to the direct surroundings of the study site. We think it is more relevant to focus on the Zugspitze summit area than to show the Zugspitze in the context of the European Alps.

[Figure]

**Figure 3: (a) Left: Location of the study site in the Zugspitze summit area. The study site is located 60 m below the peak (2963 m a.s.l.), highlighted by the red box. The hillshade is calculated from a digital elevation model with a cell size of 10 m for Austria and 5 m for Germany (Bavarian Agency for Digitisation, High-Speed Internet and Surveying, 2006). Coordinates are given in ETRS 1989 UTM Zone 32N. Right: View from above down on the study site, including dolines, the assumed main shear zones of the upper part of the rockslide, the profile used for the numerical model and the transect for the electrical resistivity survey. (b) Network of geophysical and geotechnical measurements at the study site including reference to geomorphologic and anthropogenic features. The orthofoto is derived by UAV-based photogrammetry. (c) and (d) Shear zone SZ1 filled with ice or with fine material, exposed at the inner wall of dolines. (e) Intersecting main shear zones which delineate the unstable rock mass at the south-face of the Zugspitze summit crest: without (left) and with the potential failure volume (right)."**

**RC:** "l. 161: You give "results" in the setting chapter…"

**AR:** The setting chapter (Sect. 2.1) presents the study site and the various input parameters for the numerical model. As the majority of these parameters have been measured or collected in the course of this study, they are described as results.
* * *
**RC:** "l. 177: all 2.4. is a result of investigations done for this manuscript (?)"

**AR:** Yes, the new Sect. 2.1.3 (old Sect. 2.4) presents the results of investigations performed for this study.
* * *
**RC:** "l. 225: 2.5. is a new part, and should probably be handled as such."

**AR:** The old Sect. 2.5 (new Sect. 2.1.4) contains a series of mechanical laboratory experiments to provide material parameters for the model. Therefore, the section forms part of the characterisation of the thermal and mechanical setting of the rock slope to be modelled.
* * *
**RC:** "l. 339: Why is this an own chapter? This is part of the modelling?"

**AR:** Do you refer to the old Sect. 2.5? Since this is still one part of the chapter in which all input parameters for the model are investigated and collected, it does not contain to the presentation of the model setup or the modelling procedure. For a simplification of the manuscript´s structure, we removed the titles of the two subsections

*"2.5.1 Frozen and unfrozen intact rock and rock mass"* and

*"2.5.2 Frozen and unfrozen rock joints"*

,and assigned the text to the section "*Mechanical properties*".
* * *
**RC:** "l. 454: Avoid all sentences starting "as mentioned above…"

**AR:** As proposed by the referee we deleted "as mentioned above" (l. 432):

*"We also remodelled the stability for 12 different dip angles of the fracture network with reference to the slope."*
* * *
**RC:** "l. 557, discussion: This all start is a totally uncecessary summary. You can delete the whole page and start with 5.1."

**AR:** We deleted this page so that the discussion starts after a short introduction (l. 524):

*"5. Discussion*

*This study successfully simulated the mechanical response of a warming and thawing permafrost rock slope using a temperature-dependent numerical model. The following sections show the general limitations of the presented models and critically discuss the site specific model, the model of the generalised permafrost rock slope and its transferability to other high mountain conditions.*

*5.1 Limitations*

*[…]"*

The second and third paragraph of the old page 31 were shifted to the beginning of the discussion sections 5.2 (l. 574) and 5.3 (l. 617).
* * *
**RC:** "652: The discussion should probably start with the limitations etc. It destroys a bit the flow of the discussion."

**AR:** We shifted the chapter with the limitations to the beginning of the discussion. The new structure looks as follows:

5. Discussion

>   5.1 Limitations

>   5.2 Stability assessment of the warming/thawing Zugspitze summit crest based on the factor of safety

>   5.3 The stability of a simplified permafrost rock slope with rising temperature

>   5.4 Transferability and implications for other field sites
* * *
**RC:** "l. 725: The conclusions are far too long. 50% of the conclusions is a summary, in addition to some discussion points (e.g. l. 767). You can avoid repeating stuff and citing literature in the conclusion. You should show what your study has come up with."

**AR:** Considering the suggestions of the referee, we removed references, discussion or summarising points from the conclusion. The revised conclusion looks as follows (l. 670):

See author response above.

**Response by the authors to the evaluations of referee 2:**

Dear anonymous referee,

we thank you very much for your constructive and valuable suggestions and comments related to the content of the submitted manuscript. We carefully addressed all of them and listed the changes in detail hereafter.

Philipp Mamot, on behalf of all co-authors

RC = Referee comment

AR = Author response
* * *
**RC:** "The modelling approach used in this study is an uncoupled approach and thus the title and text are misleading. Regions with anticipated temperature and rock property changes are simply assumed and not modelled. While adjusting rock properties based on temperature changes is an appropriate assumption, a thermo-mechanical coupled approach (as claimed in the title and many times in the text) also requires modeling temperature propagation." Transient temperature and rock property changes may lead to completely different results and may also modify the hydraulic situation. A strongly simplified approach, as used in this paper, needs to be confirmed by truly thermomechanical coupled numerical simulations. The required confirmation has not been provided. Thus, the validity of the approach is highly questionable."

**AR:** We interpret the term "coupled thermo-mechanical model" differently. We refer to the fact that frozen and unfrozen bedrock material properties are included, assigned to specific regions in the model and changed due to warming. However, we acknowledge that the wording can be misinterpreted. Therefore, we renamed the title into

*"A temperature-dependent mechanical model to assess the stability of degrading permafrost rock slopes".*

Accordingly, we also adjusted the wording in the text, for example (l. 70)..

*"However, no temperature-dependent mechanical numerical model has been developed yet which implements the deformation and strength reduction of permafrost bedrock and ice-filled discontinuities upon warming or thawing."*

…or (l. 647)..

*"The general procedure for a temperature-dependent mechanical stability model (Fig. 1) can be applied to any warming or thawing permafrost rock slope."*

The presented general approach for the set-up of the model and the modelling procedure (old Fig. 1) is formulated very generally. It can be easily applied to any degrading permafrost rock wall across the globe. However, consider that the model is temperature-dependent and not a

coupled one, and hence, does not require numerical thermal modelling. We proposed one specific way how this general approach can be applied to the rock slope at the Zugspitze summit crest. For instance, the conversion of the intact rock properties to those of the rock mass can also be performed by other techniques, like the Q-value.

The presented model is the first numerical attempt to assess the mechanical response of a permafrost-affected rock slope to warming/thawing. As a consequence, it is impossible to account for all relevant input parameters and possible influences in the first study/article. It is certainly necessary to include hydrostatic pressures or a heat flow model, and to perform more laboratory tests on mechanical properties in the future. A short outlook is given at the end of the conclusions (l. 710):

*"…We are at the beginning of developing numerical models for the mechanical response of a rock mass to warming/thawing. Therefore, there is a huge potential for improvements. For instance, more mechanical laboratory tests are necessary to improve process understanding and to extend the number of frozen and unfrozen material parameters for modelling. Future numerical models should be based on further sensitivity analyses, or include water pressure or the creep of ice at lower displacement rates within the advanced stage of rock slope destabilisation. Time-dependent cycling is not yet implemented in most of the numerical models, neither it is in the presented ones. Hence, they are not suited to forecast a precise time of rock slope failure. Further, the spatial warming pattern for the Zugspitze model is simplified and based on a static temperature field derived from ERT, rock thermistors and published thermal models. For a more accurate characterisation of the warming pattern, future mechanical models should include a subsurface temperature field based on a combination of geophysics and a heat flow model for different time steps."*
* * *
**RC:** "The black line in Figure 2b indicates the modelling cross section. The cross section ends in the steep rock slope south of the ridge. Thus, the modelling domain is certainly too small and boundary effects must be anticipated. A validation of the size of the model domain is not provided, but mandatory."

**AR:** We analysed the potential effect of a bigger model domain size due to additional model runs, and added a new paragraph in the list of limitations in Sect. 5.1 (Discussion) to account for this (l. 554):

*"-      The size of the Zugspitze model domain was constrained to a local section of the rock slope ending in a steep part at the south-face. The model domain fully incorporates the unstable rock mass under investigation and thereby perfectly accounts for the simulated spatial conditions of the rock-ice-mechanical processes. To investigate the slope angle at the domain boundary, we rerun the model with a bigger domain size ending in a flatter section of the south-slope (Fig. S6): the results demonstrated that a change in the size of the model domain does not significantly change the overall stability of the slope."*

The corresponding information and Fig. S6 in the supplements are as follows (l. 294):

*"To analyse the influence of a higher disturbance factor D on the stability of the Zugspitze summit crest, D was changed to a maximum of 1. Accordingly, the mechanical parameters G, K, σtm, kn and ks reduced by a mean of 56 %. While the model results showed higher displacements, the factor of safety remained unchanged (Fig. S6). Further, we extended the model domain by 20 m to the right/south and by 20 m downward to test if the model results are affected by a bigger model domain which ends in a flatter slope (Domain B in Fig. S6).*

*Again, the model results demonstrated that the overall stability of the slope does not change, although the displacements are higher by a factor of 1 to 3.*

[Figure]

| | | Principal modelling steps | Initial state | Progressive destruction of rock bridges | | | Stepwise warming from surface to core | | | | |
|---|---|---|---|---|---|---|---|---|---|---|---|
| | | Model stage number | 1 | 2 | 3 | 4 | 5 | 6 | 7 | 8 | 9 |
| Max displ. [mm] | Model domain size | A (D = 0) | 7,6 | 7,8 | 7,8 | 7,7 | 7,7 | 7,7 | 7,8 | 7,9 | 8,2 |
| | | B | 23,8 | 24,0 | 24,0 | 26,1 | 26,2 | 26,2 | 26,1 | 26,2 | 27,3 |
| | Disturbance factor | D = 1 | 18,2 | 19,5 | 19,5 | 19,4 | 19,4 | 19,5 | 17,9 | 18,9 | 19,1 |
| FoS | Model domain size | A (D = 0) | 10,9 | | | 2,6 | 3,4 | | | | 1,3 |
| | | B | 9,1 | | 4,6 | 2,2 | 2,8 | | | | 1,1 |
| | Disturbance factor | D = 1 | 10,7 | | | 2,6 | 3,4 | | | | 1,3 |

| | | Principal modelling steps | Stepwise thawing from surface to core | | | | | All unfrozen |
|---|---|---|---|---|---|---|---|---|
| | | Model stage number | 10 | 11 | 12 | 13 | 14 | 15 |
| Max displ. [mm] | Model domain size | A (D = 0) | 141,1 | 419,5 | 520,4 | 616,8 | 689,7 | 776,3 |
| | | B | 107,3 | 316,7 | 671,0 | 1066,4 | 1192,0 | 1226,4 |
| | Disturbance factor | D = 1 | 273,2 | 782,3 | 1085,0 | 1232,7 | 1327,0 | 1407,7 |
| FoS | Model domain size | A (D = 0) | 0,6 | 0,6 | | | | 0,6 |
| | | B | 0,5 | 0,6 | | | | 0,7 |
| | Disturbance factor | D = 1 | 0,6 | 0,7 | | | | 0,7 |

**Figure S6: Influence of a bigger model domain and a higher disturbance factor D on the slope stability. Model domain A and D = 0 were used for the original Zugspitze model (Section 2). The topography of the Zugspitze (red line) was extracted from a digital elevation model in ArcGIS.**

…"
* * *
**RC:** "The kinematic analysis suggests a combined planar sliding and wedge failure mode. What is dominant? Wedge failure cannot be analyzed in 2D models. Consequently, the results are unreliable."

**AR:** We added a comment in the text to make this point clear (l. 159):

*"The instability is driven by a complex combination of a plane and a wedge failure, though planar sliding is the dominant kinematic failure mode which affects the major left part of the unstable rock mass (Fig. 1a, Fig. 1e, Fig. S1). Planar sliding mainly occurs along shear zone SZ1, while wedge failure supports the displacement along SZ1/SZ3 including a tension crack SZ2 in the upper part of the slope instability. Two further shear zones (SZ4 and SZ5) form the assumed downslope boundary of the unstable rock mass in the lower slope sections. Planar*

*sliding can occur along SZ4 and SZ5, while wedge failure potentially enhances the failure process along SZ3 and the planes SZ4 and SZ5.*

*The geophysical transect and the numerical model profile run along the stepped planar failure part of the unstable rock mass (shear zones SZ1, SZ4, SZ5; left part in Fig. 1e). Shear zone SZ1 (equivalent to K4) and shear zone SZ4 (corresponds to K1) were included in the numerical model (Fig. 3b) to allow simplification and a representation in 2D. We assume the slope instability to be significantly dominated by shear zone SZ1…"*

Note, the new Fig. S1 (supplements, l. 85) is identical to the old Fig. S3. However, the new Fig. 1 looks as follows (l. 108):

[Figure]

**Figure 4: (a) Left: Location of the study site (red box) in the Zugspitze summit area, ca. 60 m below the peak (2963 m a.s.l.). The red arrow points to the sliding direction of the unstable rock mass. The hillshade is calculated from a digital elevation model with a cell size of 10 m for Austria and 5 m for Germany (Bavarian Agency for Digitisation, High-Speed Internet and Surveying, 2006). Coordinates are given in ETRS 1989 UTM Zone 32N. Right: View from above down on the study site, including dolines, the assumed main shear zones of the upper part of the rockslide, the profile used for the numerical model and the transect for the electrical resistivity survey. (b) Network of geophysical and geotechnical measurements at the study site including reference to geomorphologic and anthropogenic features. The orthofoto is**

derived by UAV-based photogrammetry. (c) and (d) Shear zone SZ1 filled with ice or with fine material, exposed at the inner wall of dolines. (e) Intersecting main shear zones which delineate the unstable rock mass at the south-face of the Zugspitze summit crest: without (left) and with the potential failure volume (right).

…"
* * *
**RC:** "The term "fracture displacement" is used in both the main body of the paper and the supplementary material. From the text it seems that linear strains (fracture opening) have been measured and no displacements (neither relative nor absolute displacements)."

**AR:** No major openings have been observed along the fractures, so the major measured elongation of crackmeters went into displacements.
* * *
**RC:** "The factor "D" in the GSI approach was set to zero, because no blasting was performed. Although the GSI approach is highly uncertain, it is clearly stated that in unstable rock slopes "D" equals 1."

**AR:** In spite of the referee´s suggestion to implement the factor D = 1, we decided to stick with D = 0. To comment on this, we included a new short discussion point in Sect. 5.1 – Limitations (l. 542):

*"[…] Fourth, a standardised recommendation for the disturbance factor D for an unstable rock slope (without blast damage) does not exist; such a recommendation should be validated numerically and include a classification of various degrees of instability. Therefore, D was set to 0, but the potential influence of a higher D on the slope stability was analysed due to additional model runs with the geometry of the Zugspitze crest (Fig. S6); a modified D approaching 1 leads to higher displacements but the factor of safety remains unchanged."*

Fig. S6 and the related information are presented in another author response above.
* * *
**RC:** „Many rock properties are derived indirectly, although laboratory tests have been performed, which allow a direct determination of static rock properties (e.g. Young's modulus and Poisson`s ratio, the normal/shear stiffness and shear resistance of fractures)."

**AR:** The determination of the joint normal/shear stiffness and the joint residual friction angle require own complex/extensive laboratory studies which are definitely beyond the scope of the presented article. The shear resistance of ice-filled joints was tested in the laboratory studies by Mamot et al. (2018, 2020). The Young´s modulus and Poisson`s ratio could have been derived directly as static properties. However, we used another common and well introduced standard technique (indirect, non-destructive) to derive them.
* * *
**RC:** "The primary aim of the model is on stability considerations, where elastic properties are typically irrelevant. This can easily be shown in a simple numerical study. Instead, assumptions are made for defining elastic properties, which are not valid. For example, it is assumed that

the ratio E/G is analogous to kn/ks. There is no theoretical justification for such an assumption (this is also stated in the cited paper)."

**AR:** It is true the elastic properties are not the most important values for the presented numerical calculation, but as we had laboratory values for them we decided to include them (since they are required as input parameters in UDEC).

The assumption for the derivation of the joint normal and shear stiffness kn and ks (with reference to the paper of Kulatilake et al., 1992) was removed from the text. Further, the text was modified as follows (l. 299):

*"…In UDEC, the deformability and strength of rock discontinuities were simulated by the Mohr-Coulomb area contact constitutive model. Rock joint parameters for ice-free and ice-filled joints were considered in the presented stability analyses. The deformability of the joints is described by the joint normal stiffness $k_n$:*

$$k_n \, [MPa/m] = \frac{E_m * E}{s * (E - E_m)}$$     *Eq. (6)*

*and the joint shear stiffness $k_s$:*

$$k_s \, [MPa/m] = \frac{G_m * G}{s * (G - G_m)}$$     *Eq. (7)*

*where s is the joint spacing (m), $E_m$ and E are the Young's moduli (GPa) for rock mass and intact rock respectively (Barton, 1972) and $G_m$ and G are the shear moduli (GPa) for rock mass and intact rock respectively (Glamheden and Lindblom, 2002; Itasca Consulting Group, 2019).*

The determination of kn and ks is consistent with Barton (1972) and the official UDEC manual. The presented equation for ks was, for instance, also used in the study by Glamheden and Lindblom (2002).
* * *
**RC:** *"*Saturation plays a major role on how the rock properties changes with changing temperature. I strongly doubt that assuming saturated rock mass conditions in a steep alpine ridge is valid. The natural saturation degree should be measured before performing laboratory tests on frozen samples and should be documented for all test specimens. The influence of the saturation degree and loading rate on the rock properties needs to be analyzed."

**AR:** The paragraph on the saturation degree of the tested rock samples (in the supplementary material, Sect. 6.1.1 – Preparation of the rock samples, l. 180) was modified to comment on this point:

*"We assume the rock mass of a real-world rock slope to be close to a saturated state. Therefore, we tested the rock samples in a frozen saturated and unfrozen saturated condition. For this, they were kept in a water bath for at least 48 h (DIN EN ISO 13755, 2002). The samples were regarded as nearly saturated when successive mass determinations yielded values varying less than 0.1 %. Frozen conditions were achieved by freezing saturated rock cores at -28 °C in a cooling box for at least 48 h.*

*A natural rock slope is expected to be nearly water-saturated for the following reasons: (i) While near surface rock saturation fluctuates highly due to meteorological influences like precipitation, wind or insulation, the rock moisture in depths greater than 15 cm remains unchanged; here, it ranges between approximately 75 and 90 % dependent on the rock type and its porosity (Rode et al., 2016; Sass, 2005). (ii) The successful application of ER surveying in permafrost bedrock in the presented study (Section 2.1.3) and in many others confirm that the investigated rock slopes are saturated, since this technique only works well in saturated*

*rock: (a) In rocks with a high degree of water saturation, electric current can be propagated due to electrolytic conduction. The DC conductivity of porous or fractured subsurface matter highly depends on saturation of the pore space and conductivity of the pore fluid (Supper et al., 2014; Telford et al., 1990). (b) The detected large ranges in electrical resistivity indicate a high degree of water saturation, since only water-saturated rocks can show a significant difference in conductivity/electrical resistivity when switching from the unfrozen to the frozen state, or vice versa (e.g. Mellor, 1973). Accordingly, water-saturated rock samples are recommended to be used for the laboratory calibration of the electrical resistivity of frozen and unfrozen rock for ERT in rock walls (Krautblatter et al., 2010)."*

---

## Author Response (AR2)

**Response by the authors:**

Dear Andreas Lang (handling editor), dear anonymous referee,

we are very glad about the positive comments/evaluation of the referee and the following decision of you, Andreas Lang, which allows a publication of our submitted and revised article after further minor corrections.

We thank the referee for his/her helpful and constructive suggestions. We carefully addressed all of them and listed the changes done in the manuscript in detail hereafter.

Enclosed you find the revised main article with and without track changes.

Kind regards

Philipp Mamot

On behalf of all co-authors

RC = Referee comment

AR = Author response
* * *
**RC:** "p. 8, 2.1.1.: Is this one sentence necessary here? Could be figure text or the 1. Sentence of 2.1.2."

**AR:** We merged the sections 2.1.1 and 2.1.2 and shifted the two sentences of the old Sect. 2.1.1 to the beginning of the new section. The new title includes both old titles. The changed manuscript looks as follows (line 131):

*"2.1 Characterisation of the mechanical and thermal setting*

*2.1.1 Model profile, rock joint geometry and kinematic analysis*

*The numerical model was set up in 2D and, thus, required a cross section of the summit crest. This profile covers a distance of 100 m, it runs from the north-face, across the crestline, to the south-face, and it crosses one of the main shear zones of the rockslide (Fig. 1a, Fig. 1b).*

*Four discontinuity sets (K1, K2, K3, K4) were identified in scan lines and field mapping (Fig. 3a). The profile for the mechanical model strikes at 146° which is in the direction of the assumed movement and follows the dip of the southern slope face (45/160). K2 (33/063) was excluded from the numerical analysis as the dip direction deviated by 66.5° from the model profile. The remaining joint sets deviating by 20–33° were implemented in the model as the standard deviations of their dip directions ranged between 5–20°, falling within the tolerable range of 30° deviation (Table 2). Joint set K1 represents the bedding planes and daylights in the south-face at an angle of 24° (Fig. 3b). .. "*

Correspondingly, references to the subsections of 2.1 in the main text were adjusted, and in Fig. 1 the reference to the section for the "Joint set geometry" was changed from 2.1.2 to 2.1.1 (line 126, see below):

[Figure]

**RC:** "Discussion: Make sure that no results are duplicated there, focus on the discussion of the results, not the results itself. E.g. p. 32, chapter 5.3. starts very descriptive. Have a look with this in mind."

**AR:** As suggested by the referee, we revised the first paragraphs of Sections 5.2 and 5.3 in the following way:

Line 585:

**"5.2 Stability assessment of the warming/thawing Zugspitze summit crest based on the factor of safety**

*The numerical model for the Zugspitze summit ridge simulates the strength reduction at warming and thawing based on laboratory tests of this and of previous studies (Table 1). Warming of ice-filled joints takes the slope close to a critical level of stability (FS = 1.3), but it does not initiate instability. As soon as thawing sets in and the ice is lost, the south-face becomes massively destabilised as displacements increase by two orders of magnitude and*

*the factor of safety falls below unity. The reduction in joint stability is significantly influenced by a loss of the joint cohesion from 0.18 MPa (at -0.5 °C) to 0 MPa (Table 5), even though the joint friction angle is increased from 13.5 to 29.3°. In contrast to our numerical analysis, Davies et al. (2001) postulate that the FS of a permafrost affected rock slope is higher for unfrozen joints than for ice-filled joints between approximately -1.5 °C and zero degrees. The differing observations on the stability upon thaw may be explained as follows: …"*

*[…]*

Line 627:

**5.3 The stability of a simplified permafrost rock slope with rising temperature**

*A permafrost rock slope with ice-filled joints and a fracture network and rock/joint properties similar to the Zugspitze south-face can become unstable at a slope gradient > 50 or 55° (transition from Domain 1 to Domain 2 in Fig. 8). This range of inclinations seems to be the most critical in terms of slope destabilisation, as it is characterised by the highest relative increase in displacements per degree of the slope angle. Interestingly, this could be confirmed by the study of the critical slope angle for varying orientations of the fracture network (Fig. 9): 50 % of the studied orientations have a critical slope angle of 50 or 55°, which corresponds to the transition from Domain 1 to Domain 2. 30 % of the orientations lead to instability with angles of between 57.5–62°, and 20 % remain stable within the studied range of slope gradients. .."*
* * *
**RC:** "Conclusions: many points, consider to merge some of them. I find the passage from l. 710 much a discussion again. Maybe stop after the first sentence as a final remark, all other stuff is could be deleted or mentioned in the discussion."

**AR:** As proposed by the referee, we merged some of the listed points in the conclusions which looks as follows (line 684):

*"[…] The related main findings are summarised as follows:*

(i) *The proposed instruction for the temperature-dependent mechanical stability model can be used for any permafrost-affected rock slope across the globe which is subjected to climatic warming.*

(ii) *Laboratory tests and field reconnaissance of the benchmark site Zugspitze exemplify thermal, geometrical and mechanical input data for the numerical model. Frozen and unfrozen bedrock material properties were assigned to specific sections in the model and changed due to warming/thawing. The modelling procedure was divided into three stages: rock bridge destruction, warming with ice-filled joints, and thawing. Process-specific and temperature-dependent input parameters were modified when switching from one stage to the next.*

(iii) *The Zugspitze model demonstrates a stability decrease towards a critical level as a result of (a) rock bridge destruction and (b) gradual warming of frozen rock and ice-filled joints from -4 to -0.5 °C. Surficial rock slope failure starts coincident to thawing of the outermost rock layer. Upon full thaw of the summit crest, expected within the next five decades, the model predicts an increase in displacements which potentially lead to final slope failure.*

*(iv)  We developed a framework to generalise and upscale the Zugspitze model. A sensitivity analysis with simplified geometry and warming pattern was performed to calculate the following critical stability thresholds: (a) The dependence between instability and the slope angle can be classified into a stable first domain (≤ 50°), a second domain with a first onset of instability (55–62°), and a third domain characterised by an accelerated slope destabilisation (≥ 64°). The greatest relative increase in displacements is observed in the second domain. (b) Warming from -4 °C to a temperature between -3 and -0.5 °C initiates instability for rock slopes ≥ 50°. Destabilisation is more pronounced for warming closer to the melting point (from -2 to -0.5 °C) than for warming from -4 to -2 °C. This difference becomes greatest in the second domain. (c) For anaclinal slopes, the critical slope angles range between 50–62°, and for cataclinal slopes, they range between 55–62°.*

*(v)  The calculated critical slope angles and rock mass temperatures correspond well to the characteristics of documented rock slope failures in permafrost areas in the European Alps, which often showed large amounts of residual ice in their scars.*

*(vi)  The critical thresholds can be applied to warming permafrost rock slopes with (a) ice-filled joints, (b) limestone equivalent to Wetterstein limestone, or probably most of the rock types relevant for permafrost rock slope failure in the Alps, (c) slope angles smaller than 70° and (d) various orientations of the fracture network consisting of three joint sets.*

*(vii)  The critical thresholds can be used to detect rock slopes which are susceptible to fail in the future and potentially endanger human life and mountain infrastructure. For this, it is a prerequisite to have data on the fracture network, lithology, geometry and the thermal field of the investigated rock slopes. In contrast, a detailed stability assessment of a single rock slope requires a number of further site-specific input data.*

*[…].."*

The last paragraph was fully shifted to the end of Section 5.1 – Limitations (line 565):

*"-  Warming to the next degree centigrade was, by default, calculated with 3000 cycles which was a compromise between a representative number of cycles for potentially reaching a numerical equilibrium (for stable rock slopes), and a reasonable, not too time-consuming calculation. Computations for low/intermediate slope angles and temperatures reached an equilibrium prior to 3000 cycles, while calculations for higher slope angles and temperatures mostly failed to reach an equilibrium (for unstable rock slopes) or scarcely required > 3000 cycles. Hence, the numerical calculation was repeated with warming steps with 6000 cycles to test a potential cut-off effect of cycles required for the rock slope to react to a change in stress-strain due to a modification of material parameters. However, the results coincide well with the model runs with 3000 cycles since displacements significantly accelerate above slope angles of 50° and 62°, and they are higher for temperatures closer to melting (Fig. S8).*

*We are at the beginning of developing numerical models for the mechanical response of a rock mass to warming/thawing. Therefore, there is a huge potential for improvements. For instance, more mechanical laboratory tests are necessary to improve process understanding and to extend the number of frozen and unfrozen material parameters for modelling. Future numerical models should be based on further sensitivity analyses, or include water pressure or the creep of ice at lower displacement rates within the advanced stage of rock slope destabilisation. Time-dependent cycling is not yet implemented in most of the numerical models, neither it is in the presented ones. Hence, they are not suited to forecast a precise time of rock slope failure. Further, the spatial warming pattern for the Zugspitze model is simplified and based on*

*a static temperature field derived from ERT, rock thermistors and published thermal models. For a more accurate characterisation of the warming pattern, future mechanical models should include a subsurface temperature field based on a combination of geophysics and a heat flow model for different time steps."*

*[…]*
* * *
**RC:** "References: This paper was in revision a very long time, make a last check that all relevant new literature is cited (little/0? references after 2017). Maybe there are none, but do a thorough check before publishing."

**AR:** We added the following new references to the main text (highlighted in red colour):

- Line 39: "*Some prominent examples are the 2003 Matterhorn block fall (0.002 Mio m³; Weber et al., 2019), the 2014 Piz Kesch rock slope failure (0.15 Mio m³; Phillips et al., 2017) and the 2017 Pizzo Cengalo failure with 8 fatalities (3–4 Mio m³; Walter et al., 2020; Mergili et al., 2020) in Switzerland, the 1987 Val Pola debris avalanche in the Italian Alps (33 Mio m³; Dramis et al., 1995), the 2005 Mt. Steller rock-ice avalanche in Alaska (40–60 Mio m³; Huggel et al., 2010) and the 2002 Kolka/Karmadon rock-ice avalanche with 140 fatalities in the Russian Caucasus (100 Mio m³; Huggel et al., 2005).*"
- Line 50: "*..The ductile temperature- and stress-dependent creep of ice and ice-rich soils has been investigated by e.g. Arenson and Springman (2005), Bray (2013) and Sanderson (1988).*"
- Line 54: "*The mechanics of frozen and unfrozen intact rock have been studied by e.g. Dwivedi et al. (2000), Inada and Yokota (1984), Kodama et al. (2013), Mellor (1973), Pläsken et al. (2020) and Voigtländer et al. (2014).*"
- Line 188: "*This technique has been used to characterise and monitor the spatial variability and evolution of mountain bedrock permafrost in steep rock walls (Keuschnig et al., 2017; Krautblatter et al., 2010; Magnin et al., 2015; Murton et al., 2016; Scandroglio et al., 2021).*"
- Line 265: "*Values for the elastic rock mass shear and bulk moduli Gm and Km were derived according to the equations presented by Tipler and Mosca (2015) for intact rock:…*"

- Line 289: "**Table 1: Laboratory-tested strength reduction of intact dolomised Wetterstein limestone due to thawing. Standard deviations (indicated with ±) are given for measured parameters, and they were used for determination of minimum and maximum values (given in parentheses) of the calculated parameters. RMC refers to parameters that are used for rock mass characterisation.**

| Mechanical parameter | Saturated frozen (-5 °C) | Saturated unfrozen (+22 °C) | Decrease due to thawing % | % °C⁻¹ | Test / equation applied | RMC |
|---|---|---|---|---|---|---|
| Density $\rho$ [g/cm³] | -- | 2.7 ± 0.01 | -- | -- | Weighing tests in water bath | x |
| Porosity $n$ [%] | -- | 0.9 ± 0.4 | -- | -- | Weighing tests in water bath | |
| Shear modulus $G$ [GPa] | 32.4 (31.7/32.6) | 25.8 (21.9/29.2) | 20.4 | 0.8 | Eq. (2), after Tipler and Mosca (2015) | |
| Bulk modulus $K$ [GPa] | 70.3 (64.8/74.8) | 55.9 (40.4/76.0) | 20.5 | 0.8 | Eq. (3), after Tipler and Mosca (2015) | |
| Young's modulus $E$ [GPa] | 84.3 (81.7/85.3) | 67.1 (55.7/77.6) | 20.4 | 0.8 | Eq. (4), after Tipler and Mosca (2015) | |
| Poisson's ratio $\upsilon$ | 0.3 ± 0.01 | 0.3 ± 0.03 | 0 | 0 | Ultrasonic tests | x |
| Dilatational wave velocity $V_D$ [m/s] | 5560 ± 50 | 4950 ± 400 | 11.0 | 0.4 | Ultrasonic tests | |

| Uniaxial tensile strength $\sigma_t$ [MPa] | 9.0 ± 1.4 | 7.2 ± 1.9 | 20.0 | 0.7 | Brazilian tests | |
|---|---|---|---|---|---|---|
| Uniaxial compressive strength $\sigma_c$ [MPa] | 109 ± 25 | 91 ± 27 | 16.5 | 0.6 | Uniaxial compressive strength tests | x |

- Line 293: **Table 2: Estimated and calculated strength reduction of Wetterstein limestone rock mass due to thawing, derived from the GSI scheme after Hoek and Brown (1997). Minimum and maximum values of calculated parameters are given in parentheses. These were determined with the standard deviations of the measured parameters (Table 3). IP = used as input parameter for the numerical model.**

| Mechanical parameter | Saturated frozen (-5 °C) | Saturated unfrozen (+22 °C) | Decrease due to thawing % | % °C$^{-1}$ | Test / equation applied | IP |
|---|---|---|---|---|---|---|
| Young's modulus $E_m$ [GPa] | 24.8 (21.7/27.5) | 22.6 (19.0/25.8) | 8.9 | 0.3 | Eq. (1), after Hoek et al. (2002) | |
| Shear modulus $G_m$ [GPa] | 9.5 (8.4/10.5) | 8.7 (7.5/9.7) | 8.4 | 0.3 | Eq. (2), after Tipler and Mosca (2015) | x |
| Bulk modulus $K_m$ [GPa] | 20.6 (17.2/24.1) | 18.9 (13.7/25.3) | 8.3 | 0.3 | Eq. (3), after Tipler and Mosca (2015) | x |
| Uniaxial tensile strength $\sigma_{tm}$ [MPa] | -0.9 (0.7/-1.1) | -0.7 (-0.5/-0.9) | 22.2 | 0.8 | Eq. (5), after Hoek et al. (2002) | x |
| Friction angle $\varphi_m$ [°] | 44[a] | 44 | 0 | 0 | estimated after Cai et al. (2004) | x |
| Cohesion $c_m$ [MPa] | 3.9 | 3.3 | 15.4 | 0.6 | estimated after Cai et al. (2004) | x |
| Parameter of the Hoek-Brown strength criterion | | | | | | |
| GSI value | -- | 65 | -- | -- | estimated after Marinos and Hoek (2000) | |
| $m_i$ | -- | 9 | -- | -- | estimated after Marinos and Hoek (2000) | |
| $m_b$ | -- | 2.6 | -- | -- | calculated after Hoek et al. (2002) | |
| $s$ | -- | 0.02 | -- | -- | calculated after Hoek et al. (2002) | |
| Disturbance factor $D$ | -- | 0 | -- | -- | estimated after Hoek et al. (2002) | |

**Note: [a]The frozen rock mass friction angle was given the same value as for the unfrozen friction angle, resulting in a decrease in thawing of zero.**

- Line 332: "*Similarly, the measured unfrozen φb corresponds well to the values listed e.g. in Barton and Choubey (1977) or Ulusay and Karakul (2016).*"

The citations have been included in the references at the end of the manuscript.